# Tighter Regret Lower Bound for Gaussian Process Bandits with Squared Exponential Kernel in Hypersphere

**Shogo Iwazaki** [1]

## Abstract

We study an algorithm-independent, worst-case lower bound for the Gaussian process (GP) bandit problem in the frequentist setting, where the reward function is fixed and has a bounded norm in the known reproducing kernel Hilbert space (RKHS). Specifically, we focus on the squared exponential (SE) kernel, one of the most widely used kernel functions in GP bandits. One of the remaining open questions for this problem is the gap in the *dimension-dependent* logarithmic factors between upper and lower bounds. This paper partially resolves this open question under a hyperspherical input domain. We show that any algorithm suffers $\Omega(\sqrt{T(\ln T)^d(\ln \ln T)^{-d}})$ cumulative regret, where $T$ and $d$ represent the total number of steps and the dimension of the hyperspherical domain, respectively. Regarding the simple regret, we show that any algorithm requires $\Omega(\epsilon^{-2}(\ln \frac{1}{\epsilon})^d(\ln \ln \frac{1}{\epsilon})^{-d})$ time steps to find an $\epsilon$-optimal point. We also provide the improved $O((\ln T)^{d+1}(\ln \ln T)^{-d})$ upper bound on the maximum information gain for the SE kernel. Our results guarantee the optimality of the existing best algorithm up to *dimension-independent* logarithmic factors under a hyperspherical input domain.

## 1. Introduction

This paper addresses an algorithm-independent lower bound for the GP bandit problem in the frequentist setting, where the underlying unknown reward function is fixed and belongs to the known RKHS. In existing lower-bound studies, two commonly used kernels are the SE and Matérn kernels (Scarlett et al., 2017). For the Matérn kernel, the state-of-the-art upper bound achieves the optimal regret up to

[1]LY Corporation, Tokyo, Japan. Correspondence to: Shogo Iwazaki <siwazaki@lycorp.co.jp>.

*Proceedings of the 43rd International Conference on Machine Learning*, Seoul, South Korea. PMLR 306, 2026. Copyright 2026 by the author(s).

dimension-independent logarithmic factors (e.g., Camilleri et al., 2021; Li & Scarlett, 2022; Vakili et al., 2021a; Valko et al., 2013; Salgia et al., 2021). However, for the SE kernel, there remains a substantial gap between the upper and lower bounds. Specifically, the current best upper bound for the SE kernel shows that the cumulative and simple regrets are $O^*(\sqrt{T(\ln T)^d})$ and $O^*(\sqrt{(\ln T)^d/T})$, respectively. Here, as with (Scarlett et al., 2017), we use the notation $O^*(\cdot)$ to hide dimension-independent logarithmic factors. In contrast, the current best lower bounds for cumulative and simple regrets are $\Omega(\sqrt{T(\ln T)^{d/2}})$ and $\Omega(\sqrt{(\ln T)^{d/2}/T})$, respectively (Cai & Scarlett, 2021; Scarlett et al., 2017). Thus, there remains a *dimension-dependent* $(\ln T)^{d/4}$ gap between the upper and lower bounds. These dimension-dependent logarithmic factors induce a non-negligible gap between the upper and lower bounds. For example, even for moderate dimensions such as $d \leq 20$, which are commonly considered suitable for practical applications (Frazier, 2018), the resulting $(\ln T)^{d/4}$ gap can scale up to the fifth power of $(\ln T)$ in the worst case. Furthermore, many researchers explore much higher-dimensional settings (e.g., Eriksson et al., 2019; Hvarfner et al., 2024; Iwazaki et al., 2025; Kandasamy et al., 2015; Wang et al., 2016). Therefore, closing such gaps in the logarithmic factors, which grow exponentially with $d$, remains a fundamental open problem. We partially resolve this problem by showing an $\Omega(\sqrt{T(\ln T)^d(\ln \ln T)^{-d}})$ regret lower bound for a special case where the input domain is the hypersphere $\mathbb{S}^d := \{\boldsymbol{x} \in \mathbb{R}^{d+1} \mid \|\boldsymbol{x}\|_2 = 1\}$.

**Contributions.** Our contributions are as follows:

- We provide an improved, algorithm-independent worst-case lower bound for the SE kernel on a hypersphere. Specifically, under a hyperspherical input domain $\mathcal{X} = \mathbb{S}^d$, we show that any algorithm suffers $\Omega(\sqrt{T(\ln T)^d(\ln \ln T)^{-d}})$ cumulative regret. Furthermore, for the simple regret, we show that any algorithm requires $\Omega(\epsilon^{-2}(\ln \frac{1}{\epsilon})^d(\ln \ln \frac{1}{\epsilon})^{-d})$ time steps to find an $\epsilon$-optimal point. Rigorous statements are provided in Theorems 3.1 and 3.2 in Section 3. These results fill the $(\ln T)^{d/4}$ gap in the existing results. Table 1 summarizes existing results and our new lower bounds.

- From a technical perspective, a key contribution is the

*Table 1.* A comparison between best known upper bounds and lower bounds for the SE kernel on the hyperspherical input domain $\mathcal{X} = \mathbb{S}^d$. For the simple regret, the table below shows the upper and lower bounds on the number of time steps required for the simple regret to become smaller than $\epsilon > 0$.

|  | Cumulative regret | Simple regret |
|---|---|---|
| Upper bound | $O^*\left(\sqrt{T(\ln T)^d}\right)$ | $O^*\left(\frac{1}{\epsilon^2}\left(\ln\frac{1}{\epsilon}\right)^d\right)$ |
| Lower bound (Scarlett et al., 2017) | $\Omega\left(\sqrt{T(\ln T)^{d/2}}\right)$ | $\Omega\left(\frac{1}{\epsilon^2}\left(\ln\frac{1}{\epsilon}\right)^{d/2}\right)$ |
| Lower bound (**Ours**) | $\Omega\left(\sqrt{\frac{T(\ln T)^d}{(\ln\ln T)^d}}\right)$ | $\Omega\left(\frac{\left(\ln\frac{1}{\epsilon}\right)^d}{\epsilon^2\left(\ln\ln\frac{1}{\epsilon}\right)^d}\right)$ |

construction of a new hard function class based on Mercer's representation theorem and spherical harmonics theory. The details are provided in Section 4.

- We also prove an $O((\ln T)^{d+1}(\ln\ln T)^{-d})$ upper bound on the maximum information gain for the SE kernel under the hypersphere $\mathbb{S}^d$ or any compact input domain in $\mathbb{R}^d$, which provides an $O(\sqrt{(\ln\ln T)^d})$ improvement for many existing algorithms relying on the information gain-based analysis.

**Limitations.** A limitation of our results is that we cannot claim the same improved lower bounds for a general compact input domain in $\mathbb{R}^d$. That said, we believe that our analysis marks an important milestone in reconsidering the existing lower bounds for the SE kernel. In Section 6, we provide a discussion of how to extend our core idea to a compact input domain in $\mathbb{R}^d$. Nevertheless, even under this limitation, we would like to note that our result is valuable in its own right, since the hyperspherical domain is often studied in existing kernel-based or neural-based bandit algorithms (e.g., Iwazaki & Suzumura, 2024; Kassraie & Krause, 2022; Salgia, 2023; Vakili et al., 2021b; Zhou et al., 2020). Furthermore, whereas our improved lower bound is limited to the hyperspherical domain $\mathbb{S}^d$, our improved upper bound via the maximum information gain is quite general and is applicable to an arbitrary compact input domains in $\mathbb{R}^d$.

### 1.1. Related Works

**GP Bandits.** GP bandits have been extensively studied for applications with black-box reward functions, such as robotics (Martinez-Cantin et al., 2007), experimental design (Lei et al., 2021), and hyperparameter tuning (Snoek et al., 2012). Although this paper focuses on the frequentist assumption in a noisy feedback setting, alternative assumptions have also been studied, such as Bayesian settings (Iwazaki, 2025b; Russo & Van Roy, 2014; Srinivas et al., 2010) and the noise-free feedback setting (Bull, 2011; Iwazaki, 2025a; Iwazaki & Takeno, 2025a; Vakili, 2022; Xu

et al., 2024). We next summarize the literature for regret analysis under the noisy, frequentist setting, which is the setting considered in this paper.

**Regret Lower Bounds.** The first worst-case regret lower bounds were provided by Scarlett et al. (2017). They show the lower bounds for the SE and $\nu$-Matérn kernels, summarized in Table 1. Subsequently, several studies extend their ideas to obtain worst-case lower bounds for more advanced settings, e.g., robust settings (Bogunovic et al., 2018; Cai & Scarlett, 2021), heavy-tailed reward settings (Ray Chowdhury & Gopalan, 2019), and non-stationary settings (Cai & Scarlett, 2025; Iwazaki & Takeno, 2025b). However, as in the standard setting of (Scarlett et al., 2017), the lower bounds for these advanced settings also exhibit gaps in the dimension-dependent logarithmic factors between the upper and lower bounds for the SE kernel.

**Regret Upper Bounds.** Regarding cumulative regret, Srinivas et al. (2010) provide the first guarantees for the GP-based upper-confidence-bound (GP-UCB) algorithm. They show that its cumulative regret is $O(\gamma_T\sqrt{T})$, where $\gamma_T$ is a kernel-dependent complexity parameter called the *maximum information gain* (MIG). Chowdhury & Gopalan (2017) show that GP-based Thompson sampling (GP-TS) also achieves $O^*(\gamma_T\sqrt{T})$ cumulative regret. It is known that these $O^*(\gamma_T\sqrt{T})$ cumulative regrets can be improved to $O^*(\sqrt{T\gamma_T})$ using non-adaptive sampling-based algorithms (Camilleri et al., 2021; Li & Scarlett, 2022; Salgia et al., 2021; Valko et al., 2013). By supplying explicit upper bounds on the MIG, we can obtain an explicit upper bound on the current best $O^*(\sqrt{T\gamma_T})$ regret bound. For the SE and $\nu$-Matérn kernels with $\nu > 1/2$ on a compact input domain $\mathcal{X} \subset \mathbb{R}^d$, $\gamma_T = O((\ln T)^{d+1})$ and $\gamma_T = O(T^{\frac{d}{2\nu+d}}(\ln T)^{\frac{4\nu+d}{2\nu+d}})$ hold, respectively (Iwazaki, 2025b; Srinivas et al., 2010)[1]. Furthermore, when $\mathcal{X} \subset \mathbb{S}^d$, the same upper bounds hold as for compact $\mathcal{X} \subset \mathbb{R}^d$ (Appendix B in Iwazaki, 2025b). Thus, for $\mathcal{X} \subset \mathbb{R}^d$ or $\mathcal{X} \subset \mathbb{S}^d$, the current best upper bounds are $O^*(\sqrt{T(\ln T)^{d/2}})$ and $O^*\left(T^{\frac{\nu+d}{2\nu+d}}(\ln T)^{\frac{4\nu+d}{2\nu+d}}\right)$ for the SE and $\nu$-Matérn kernels, respectively. Here, for $\nu$-Matérn kernel, note that the logarithmic term satisfies $(\ln T)^{\frac{4\nu+d}{2\nu+d}} \leq (\ln T)^2$ for any $d$ (and any $\nu$); therefore, the regret for the Matérn kernel is already optimal up to dimension-independent logarithmic factors. Finally, it is known that simple regret can also be analyzed in terms of the MIG, and its current best upper bound is

---

[1] For the $\nu$-Matérn kernel, Vakili et al. (2021c) show tighter $O(T^{\frac{d}{2\nu+d}}(\ln T)^{\frac{2\nu}{2\nu+d}})$ upper bound on the MIG under uniform boundedness assumption of the eigenfunctions of the kernel. To our knowledge, the validity of this assumption for general compact input domains is unclear (e.g., Chapter 4.4 in Janz, 2022). Therefore, we refer to a weaker but more general bound from (Iwazaki, 2025b)

$O^*(\sqrt{\gamma_T/T})$ (Vakili et al., 2021a). By substituting the explicit upper bound of MIG, we can also confirm that the simple regret suffers from dimension-dependent logarithmic factors.

**Spherical Harmonics in GP Bandits.** Our proof utilizes the theory of *spherical harmonics* (Atkinson & Han, 2012; Efthimiou & Frye, 2014), which gives the explicit Mercer decomposition of a continuous kernel on the hyperspherical input domain. Several GP-based or neural-network-based bandit researches leverage spherical harmonics to obtain the upper bound on the MIG on the hypersphere (Iwazaki & Suzumura, 2024; Iwazaki, 2025b; Kassraie & Krause, 2022; Vakili et al., 2021b). Specifically, our improved upper bound on the MIG in Section 5 is built on the proof of (Iwazaki, 2025b). However, to our knowledge, no prior work uses spherical harmonics to establish lower bounds.

## 2. Preliminaries

### 2.1. Basic Problem Description

The GP bandit problem is formulated as a sequential decision-making problem, with a reward function $f$ lying in a known RKHS. Let $k : \mathcal{X} \times \mathcal{X} \to \mathbb{R}$ be a positive definite kernel function, where $\mathcal{X}$ is a compact subset of Euclidean space. Then, there exists an RKHS $\mathcal{H}_k(\mathcal{X})$, whose reproducing kernel is $k$ (Aronszajn, 1950). In GP bandits, the underlying reward function $f : \mathcal{X} \to \mathbb{R}$ is assumed to belong to $\mathcal{H}_k(\mathcal{X})$ and to have bounded RKHS norm $\|f\|_{\mathcal{H}_k(\mathcal{X})} \leq B < \infty$. Here, we also define $\mathcal{H}_k(\mathcal{X}, B) := \{f \in \mathcal{H}_k(\mathcal{X}) \mid \|f\|_{\mathcal{H}_k(\mathcal{X})} \leq B\}$ as the hypothesis function class of the problem. For the choice of the kernel, this paper focuses on the following SE kernel:

$$k(\boldsymbol{x}, \widetilde{\boldsymbol{x}}) = \exp\left(-\frac{\|\boldsymbol{x} - \widetilde{\boldsymbol{x}}\|_2^2}{\theta}\right), \qquad (1)$$

where $\theta > 0$ is the lengthscale parameter.

The learner sequentially interacts with an unknown function $f$ in the RKHS. At each step $t$, the learner chooses the query point $\boldsymbol{x}_t \in \mathcal{X}$ based on the history up to step $t - 1$. Then, the learner observes a noisy function value $y_t := f(\boldsymbol{x}_t) + \epsilon_t$, where $\epsilon_t \sim \mathcal{N}(0, \sigma^2)$ is the mean-zero Gaussian noise with variance $\sigma^2 > 0$. Here, $(\epsilon_t)_{t \in \mathbb{N}_+}$ is assumed to be independent across $t \in \mathbb{N}_+$. The learner's goal is to minimize either the cumulative regret $R_T := \sum_{t \in [T]} (\max_{\boldsymbol{x} \in \mathcal{X}} f(\boldsymbol{x}) - f(\boldsymbol{x}_t))$ or the simple regret $r_T := \max_{\boldsymbol{x} \in \mathcal{X}} f(\boldsymbol{x}) - f(\widehat{\boldsymbol{x}}_T)$, where $[T] := \{1, \ldots, T\}$, and $\widehat{\boldsymbol{x}}_T \in \mathcal{X}$ is the estimated maximizer reported by the algorithm at the end of step $T$. This paper studies the algorithm-independent lower bounds for these performance measures.

### 2.2. Summary of the Existing Proof

Our new proof is built on the existing proof strategy developed in (Cai & Scarlett, 2021; Scarlett et al., 2017), which study a worst-case lower bound for the input domain $\mathcal{X} = [0, 1]^d$. We briefly summarize their proof in this subsection. Roughly speaking, their proof can be divided into the following two steps: (i) the construction of the finite "hard" function class $\mathcal{F}_\epsilon \subset \mathcal{H}_k(\mathcal{X}, B)$ characterized by a positive parameter $\epsilon > 0$, and (ii) bounding the worst-case regret by "change of measure" arguments on $\mathcal{F}_\epsilon$. Through these two steps, the lower bound on the time steps required to find an $\epsilon$-optimal point is quantified. Regarding the cumulative regret, the lower bound as a function of the parameter $\epsilon$ is quantified; subsequently, the desired lower bound is also obtained by selecting $\epsilon > 0$ depending on $T, \sigma^2$, and $B$ so that the final lower bound becomes as high as possible.

#### 2.2.1. CONSTRUCTION OF FINITE FUNCTION CLASS

Scarlett et al. (2017) first constructed the finite function class $\mathcal{F}_\epsilon$ to show the lower bound over this class. Intuitively, to obtain a large lower bound, $\mathcal{F}_\epsilon$ should be designed so that (i) it is hard for the learner to identify the true $f \in \mathcal{F}_\epsilon$ from the noisy feedback, and (ii) the learner suffers from a certain amount of regret while identifying the true underlying function $f$. To do so, Scarlett et al. (2017) propose to construct $\mathcal{F}_\epsilon$ by arranging shifted versions of a common function $g_\epsilon$, which only attains large values around $\boldsymbol{0}$. Specifically, Scarlett et al. (2017) chooses $g_\epsilon$ based on the inverse Fourier transform $h$ of a *bump function* as follows:

$$g_\epsilon(\boldsymbol{x}) = \frac{2\epsilon}{h(\boldsymbol{0})} h\left(\frac{\zeta \boldsymbol{x}}{w_\epsilon}\right), \qquad (2)$$

where $w_\epsilon > 0$ is the parameter depending on $\epsilon$, and $\zeta > 0$ is some absolute constant. See Section III in (Scarlett et al., 2017) for a more detailed description of $h$ and $\zeta$. The following lemma specifies the properties of the function $g_\epsilon$ in Eq. (2) and summarizes Sections III and IV in (Scarlett et al., 2017).

**Lemma 2.1** (Properties of $g_\epsilon$). *Fix $d \in \mathbb{N}_+$, $\epsilon, B > 0$, and let $k : \mathbb{R}^d \times \mathbb{R}^d \to \mathbb{R}$ be the SE kernel with lengthscale parameter $\theta > 0$. Assume that $\epsilon/B$ is sufficiently small. Furthermore, set $w_\epsilon > 0$ to $w_\epsilon = \Theta\left(\ln^{-1/2}(B/\epsilon)\right)^2$. Then, $g_\epsilon : \mathbb{R}^d \to \mathbb{R}$ in Eq. (2) satisfies the following properties:*

1. *The function $g_\epsilon$ attains the maximum at $\boldsymbol{0}$ and $g_\epsilon(\boldsymbol{0}) = 2\epsilon$. Furthermore, for all $\boldsymbol{x} \in \mathbb{R}^d$, $|g_\epsilon(\boldsymbol{x})| \leq 2\epsilon$.*

---

[2]Unless otherwise stated, in the proof of the lower bounds, the hidden constant of the order notation depends only on $d$ and $\theta$. Similarly, the words "sufficiently small" and "sufficiently large" mean that there exist lower or upper threshold constants depending only on $d$ and $\theta$. For example, "the event $A$ holds if $\epsilon/B$ is sufficiently small" means that there exists a constant $\epsilon_{d,\theta} > 0$ such that $A$ holds under $0 < \epsilon/B \leq \epsilon_{d,\theta}$.

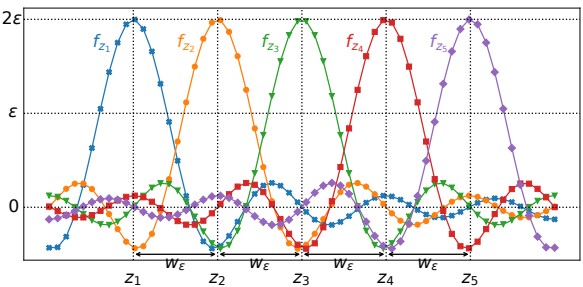

*Figure 1.* Illustrative example of $\mathcal{F}_\epsilon$ in one dimension (adapted from Figure 1 in Scarlett et al., 2017).

2. *The $\epsilon$-optimal region is a subset of the $L_\infty$-ball of radius $w_\epsilon/2$. Namely, $\{\boldsymbol{x} \in \mathbb{R}^d \mid \max_{\widetilde{\boldsymbol{x}} \in \mathbb{R}^d} g_\epsilon(\widetilde{\boldsymbol{x}}) - g_\epsilon(\boldsymbol{x}) \leq \epsilon\} \subset \{\boldsymbol{x} \in \mathbb{R}^d \mid \|\boldsymbol{x}\|_\infty \leq w_\epsilon/2\}$ holds.*

3. *The RKHS norm satisfies $\|g_\epsilon\|_{\mathcal{H}_k(\mathbb{R}^d)} \leq B$.*

Based on the above $g_\epsilon$, let us define the following finite function class $\mathcal{F}_\epsilon$.

**Definition 2.2** (Finite function class construction based on $g_\epsilon$). Let us define $M_\epsilon := \lfloor 1/w_\epsilon \rfloor^d$. Furthermore, let $\mathcal{N}_\epsilon \subset \mathcal{X}$ be uniformly spaced grid points in $\mathcal{X} := [0, 1]^d$ such that $|\mathcal{N}_\epsilon| = M_\epsilon$ and $\|\boldsymbol{z}_1 - \boldsymbol{z}_2\|_\infty > w_\epsilon$ for any two distinct points $\boldsymbol{z}_1, \boldsymbol{z}_2 \in \mathcal{N}_\epsilon$. Then, we define the function class $\mathcal{F}_\epsilon := (f_{\boldsymbol{z};\epsilon})_{\boldsymbol{z} \in \mathcal{N}_\epsilon}$, where $f_{\boldsymbol{z};\epsilon} : \mathcal{X} \to [-2\epsilon, 2\epsilon]$ is defined as $f_{\boldsymbol{z};\epsilon}(\boldsymbol{x}) = g_\epsilon(\boldsymbol{x} - \boldsymbol{z})$.

An illustrative image of $\mathcal{F}_\epsilon$ is provided in Figure 1. Before moving to the next subsection, we would like to note the following properties of $\mathcal{F}_\epsilon$, which are leveraged in the proof.

1. Since the input shift operation and restriction operation to $\mathcal{X} = [0, 1]^d$ do not increase the RKHS norm (Part I.5 in Aronszajn, 1950), we have $\mathcal{F}_\epsilon \subset \mathcal{H}_k(\mathcal{X}, B)$ for $\epsilon$ that satisfies the condition in Lemma 2.1.

2. From the definition of $\mathcal{N}_\epsilon$ and property 2 in Lemma 2.1, we can decompose $\mathcal{X} := [0, 1]^d$ into a collection of distinct regions $(\mathcal{R}_{\boldsymbol{z};\epsilon})_{\boldsymbol{z} \in \mathcal{N}_\epsilon}$ such that the $\epsilon$-optimal regions of $f_{\boldsymbol{z};\epsilon}$ are distinct. Namely, $(\mathcal{R}_{\boldsymbol{z};\epsilon})_{\boldsymbol{z} \in \mathcal{N}_\epsilon}$ is defined such that (i)$\bigsqcup_{\boldsymbol{z} \in \mathcal{N}_\epsilon} \mathcal{R}_{\boldsymbol{z};\epsilon} = \mathcal{X}$, (ii)$\boldsymbol{z} \in \mathcal{R}_{\boldsymbol{z};\epsilon}$, and (iii)$\forall \boldsymbol{z} \in \mathcal{R}_{\boldsymbol{z}_2;\epsilon}, \|\boldsymbol{z}_1 - \boldsymbol{z}\|_\infty > w_\epsilon/2$ hold for any two distinct points $\boldsymbol{z}_1, \boldsymbol{z}_2 \in \mathcal{N}_\epsilon$.

### 2.2.2. BOUNDING REGRET

The next step is to obtain the lower bound on the regret over $\mathcal{F}_\epsilon$. Intuitively, to obtain a worst-case regret within $\mathcal{F}_\epsilon$, it is desired to capture the difference in the algorithm's behavior on two distinct functions $f, \widetilde{f} \in \mathcal{F}_\epsilon$. To do so, the existing works rely on "change of measure" arguments (Cai & Scarlett, 2021; Scarlett et al., 2017), which connect the number of query points with the difference between the two

underlying measures under $f$ and $\widetilde{f}$. In this paper, we follow the proof strategy in (Cai & Scarlett, 2021) and leverage the following lemma.

**Lemma 2.3** (Relating two instances, Lemma 1 in (Cai & Scarlett, 2021)). *Let us consider the GP bandit problem defined in Section 2.1. Fix any $f, \widetilde{f} \in \mathcal{H}_k(\mathcal{X}, B)$, any $\delta \in (0, 1/3)$, and any algorithm. Let $(\mathcal{R}_j)_{j \in [M]}$ be a partition of the input space into $M$ disjoint regions, and let $\mathcal{A}$ be any event depending on the history up to step $T \in \mathbb{N}_+$. Then, if $\mathbb{P}_f(\mathcal{A}) \geq 1 - \delta$ and $\mathbb{P}_{\widetilde{f}}(\mathcal{A}) \leq \delta$, we have*

$$\sum_{j \in [M]} \mathbb{E}_f[N_j(T)] \overline{D}_{f,\widetilde{f}}^{(j)} \geq \ln \frac{1}{2.4\delta}, \qquad (3)$$

*where $N_j(T) := \sum_{t=1}^T \mathbb{1}\{\boldsymbol{x}_t \in \mathcal{R}_j\}$ is the number of query points within $\mathcal{R}_j$ up to step $T$. Furthermore,*

$$\overline{D}_{f,\widetilde{f}}^{(j)} := \sup_{\boldsymbol{x} \in \mathcal{R}_j} \mathrm{KL}\left(\mathcal{N}(f(\boldsymbol{x}), \sigma^2) \| \mathcal{N}(\widetilde{f}(\boldsymbol{x}), \sigma^2)\right) \quad (4)$$

*is the maximum Kullback–Leibler (KL) divergence from samples between $f$ and $\widetilde{f}$ in $\mathcal{R}_j$. Here, $\mathcal{N}(\mu, \sigma^2)$ denotes the Gaussian measure with the mean $\mu$ and variance $\sigma^2$.*

When we use the above lemma, we need to handle the maximum KL-divergence $\overline{D}_{f,\widetilde{f}}^{(j)}$. The following lemma is useful for bounding that term.

**Lemma 2.4** (Lemma 5 in (Scarlett et al., 2017)). *Let $\mathcal{F}_\epsilon$, $\mathcal{N}_\epsilon$, and $(\mathcal{R}_{\boldsymbol{z};\epsilon})_{\boldsymbol{z} \in \mathcal{N}_\epsilon}$ be the function class, grid points, and the decomposition of $\mathcal{X}$ defined in Section 2.2.1. Then, for all $\boldsymbol{z} \in \mathcal{N}_\epsilon$, $\sum_{\widetilde{\boldsymbol{z}} \in \mathcal{N}_\epsilon} \sup_{\boldsymbol{x} \in \mathcal{R}_{\boldsymbol{z};\epsilon}} \left(f_{\widetilde{\boldsymbol{z}};\epsilon}(\boldsymbol{x})\right)^2 \leq C_d \epsilon^2$, where $C_d > 0$ is a constant depending on $d$[3].*

The remainder of the proof applies Lemma 2.3, using the properties of $\mathcal{F}_\epsilon$ established in Lemmas 2.1 and 2.4. See the proof in (Cai & Scarlett, 2021) for details.

### 2.2.3. LOWER BOUNDS ON THE HYPERSPHERE

The aforementioned proof strategy in (Cai & Scarlett, 2021) assumes $\mathcal{X} = [0, 1]^d$. However, by extending the construction of $\mathcal{F}_\epsilon$, $\mathcal{N}_\epsilon$, and $M_\epsilon$ based on the standard packing argument, we can straightforwardly derive a domain-dependent lower bound. Specifically, we can follow the same proof strategy for general $\mathcal{X}$ by simply replacing $M_\epsilon$ and $\mathcal{N}_\epsilon$ in Definition 2.2 by the $w_\epsilon$-packing number and the corresponding $w_\epsilon$-separated set of $\mathcal{X}$, respectively. Below, we specify the lower bounds for $\mathcal{X} = \mathbb{S}^d$ as the corollaries of the main theorems in (Cai & Scarlett, 2021).

**Corollary 2.5** (Simple regret lower bound on the hypersphere, extended from (Cai & Scarlett, 2021)). *Fix any*

---

[3]Here, from the original lemma in (Scarlett et al., 2017), we omit two statements that are unnecessary for the proof in (Cai & Scarlett, 2021).

$\delta \in (0, 1/3)$, $\epsilon \in (0, 1/2)$, $B > 0$, and $T \in \mathbb{N}_+$. *Let us consider the GP bandits problem on $\mathcal{X} = \mathbb{S}^d$ with the SE kernel described in Section 2.1. Suppose that $d \in \mathbb{N}_+$ and $\theta > 0$ are fixed constants. Furthermore, suppose that there exists an algorithm that achieves $r_T \leq \epsilon$ with probability at least $1 - \delta$ for any $f \in \mathcal{H}_k(\mathcal{X}, B)$. Then, if $\epsilon/B$ is sufficiently small, it is necessary that*

$$T = \Omega\left(\frac{\sigma^2}{\epsilon^2}\left(\ln \frac{B}{\epsilon}\right)^{d/2} \ln \frac{1}{\delta}\right). \tag{5}$$

**Corollary 2.6** (Cumulative regret lower bound on the hypersphere, extended from (Cai & Scarlett, 2021)). *Consider the same setting as in Corollary 2.5. Furthermore, assume $\sigma^2(\ln(1/\delta))/B^2 = O(T)$ with sufficiently small implied constant. Then, for any algorithm, there exists a function $f \in \mathcal{H}_k(\mathcal{X}, B)$ such that the following inequality holds with probability at least $\delta$:*

$$R_T = \Omega\left(\sqrt{T\sigma^2\left(\ln \frac{B^2 T}{\sigma^2 \ln \frac{1}{\delta}}\right)^{d/2}}\right). \tag{6}$$

Proofs are given in Appendix A for completeness. The aim of this paper is to improve the lower bounds stated in Corollaries 2.5 and 2.6.

### 2.3. Summary of Mercer's Representation Theorem

Our proof leverages Mercer's representation theorem to construct a new hard function class. This provides a feature representation of the RKHS based on the eigensystem of the kernel integral operator. Given a positive definite kernel $k : \mathcal{X} \times \mathcal{X} \to \mathbb{R}$ and a finite Borel measure $\mu$, let $\mathcal{T}_k : L_2(\mathcal{X}) \to L_2(\mathcal{X})$ denote the integral operator of $k$, which is defined as $(\mathcal{T}_k f)(\cdot) = \int_{\mathcal{X}} f(\boldsymbol{x})k(\boldsymbol{x}, \cdot)\mu(d\boldsymbol{x})$. Here, $L_2(\mathcal{X}) := \{f : \mathcal{X} \to \mathbb{R} \mid \int_{\mathcal{X}} f(\boldsymbol{x})^2 \mu(d\boldsymbol{x}) < \infty\}$ is the set of square-integrable functions with respect to $\mu$. Let $(\phi_i)_{i \in \mathbb{N}}$ and $(\lambda_i)_{i \in \mathbb{N}}$ be orthonormal eigenfunctions and corresponding eigenvalues of $\mathcal{T}_k$. That is, $(\phi_i)_{i \in \mathbb{N}}$ and $(\lambda_i)_{i \in \mathbb{N}}$ satisfy $\int_{\mathcal{X}} \phi_i(\boldsymbol{x})\phi_j(\boldsymbol{x})\mu(d\boldsymbol{x}) = \mathbb{1}\{i = j\}$ and $\mathcal{T}_k \phi_i = \lambda_i \phi_i$. Then, Mercer's representation theorem provides an explicit representation of the RKHS and its norm.

**Theorem 2.7** (Mercer representation, e.g., Theorem 4.2 in (Kanagawa et al., 2018)). *Let $\mathcal{X}$ be a compact metric space, $k$ be a continuous positive definite kernel, and $\mu$ be a finite Borel measure whose support is $\mathcal{X}$. Then, $\mathcal{H}_k(\mathcal{X})$ is represented as*

$$\mathcal{H}_k(\mathcal{X}) = \left\{f := \sum_{i \in \mathbb{N}} \alpha_i \sqrt{\lambda_i} \phi_i \;\Big|\; \|f\|_{\mathcal{H}_k(\mathcal{X})}^2 := \sum_{i \in \mathbb{N}} \alpha_i^2 < \infty\right\}.$$

Our proof constructs a hard function based on the Mercer representation for the SE kernel $k = k_{\mathrm{SE}}$ on the hypersphere $\mathbb{S}^d$. Although the explicit form of the eigenfunctions

$(\phi_i)$ are hard to obtain for a general compact input domain $\mathcal{X}$, the hypersphere $\mathbb{S}^d$ is a notable exception where the eigensystems can be specified explicitly.

#### 2.3.1. SPHERICAL HARMONICS AND MERCER DECOMPOSITION ON THE HYPERSPHERE

When $\mathcal{X} = \mathbb{S}^d$, it is known that the eigensystem of the integral operator associated with the SE kernel is specified by special polynomials on $\mathbb{S}^d$, called *spherical harmonics* (Minh et al., 2006). In this subsection, we briefly summarize the known results about spherical harmonics. For the readers who are not familiar with the contents of this subsection, we refer to (Atkinson & Han, 2012; Efthimiou & Frye, 2014) as the basic textbooks of spherical harmonics. We also refer to, e.g., (Vakili et al., 2021b) and Appendix B.2 in (Iwazaki, 2025b) as the references that use spherical harmonics in the context of GP bandits.

Let us consider a polynomial $p(\cdot) : \mathbb{R}^{d+1} \to \mathbb{R}$, i.e., the function $p(\cdot)$ in $\mathbb{R}^{d+1}$ is represented as a form $p((x_1, \ldots, x_{d+1})^\top) = \sum_{\boldsymbol{\alpha} \in \mathcal{A}} c_{\boldsymbol{\alpha}} x_1^{\alpha_1} x_2^{\alpha_2} \ldots x_d^{\alpha_d} x_{d+1}^{\alpha_{d+1}}$, $\mathcal{A} := \{\boldsymbol{\alpha} := (\alpha_1, \ldots, \alpha_{d+1})^\top \mid \boldsymbol{\alpha} \in \mathbb{N}^{d+1}\}$. Here, $(c_{\boldsymbol{\alpha}})_{\boldsymbol{\alpha} \in \mathcal{A}}$ are some coefficients, where $c_{\boldsymbol{\alpha}} \in \mathbb{R}$ for any $\boldsymbol{\alpha} \in \mathcal{A}$. We call a polynomial $p(\cdot)$ a *homogeneous* polynomial of degree $n$ if the polynomial $p(\cdot)$ is of the form $p((x_1, \ldots, x_{d+1})^\top) = \sum_{\boldsymbol{\alpha}; \sum_{i=1}^{d+1} \alpha_i = n} c_{\boldsymbol{\alpha}} x_1^{\alpha_1} x_2^{\alpha_2} \ldots x_d^{\alpha_d} x_{d+1}^{\alpha_{d+1}}$. Furthermore, we call a polynomial $p(\cdot)$ *harmonic* if $\Delta p(\boldsymbol{x}) := \sum_{i=1}^{d+1} \frac{\partial^2}{\partial x_i^2} p(\boldsymbol{x}) = 0$ for all $\boldsymbol{x} \in \mathbb{R}^{d+1}$. Below, we formally define the spherical harmonics.

**Definition 2.8** (Spherical harmonics, e.g., Chapter 2.1.3 in (Atkinson & Han, 2012) or Chapter 4.2 in (Efthimiou & Frye, 2014)). Let $\mathbb{Y}_n^{d+1}(\mathbb{R}^{d+1})$ be the space that consists of all homogeneous polynomials of degree $n$ in $\mathbb{R}^{d+1}$ that are also harmonic. Furthermore, let us define $\mathbb{Y}_n^{d+1} := \mathbb{Y}_n^{d+1}(\mathbb{R}^{d+1})|_{\mathbb{S}^d}$, where $\mathbb{Y}_n^{d+1}(\mathbb{R}^{d+1})|_{\mathbb{S}^d}$ is the space of all the functions of $\mathbb{Y}_n^{d+1}(\mathbb{R}^{d+1})$ whose input domain is restricted to $\mathbb{S}^d$. Then, any element $p : \mathbb{S}^d \to \mathbb{R}$ in $\mathbb{Y}_n^{d+1}$ is called a spherical harmonics.

Below, we summarize the basic properties of spherical harmonics used in the main text:

- **Orthogonality and Dimension** (Theorem 4.6 in (Efthimiou & Frye, 2014) or Chapter 2.1.3 in (Atkinson & Han, 2012)). Let $\mu_{\mathbb{S}^d}$ be the induced Lebesgue measure on $\mathbb{S}^d$[4]. Then, for any $Y \in \mathbb{Y}_n^{d+1}$ and $\widetilde{Y} \in \mathbb{Y}_m^{d+1}$ such that $n \neq m$, we have $\int_{\mathbb{S}^d} Y(\boldsymbol{x})\widetilde{Y}(\boldsymbol{x})\mu_{\mathbb{S}^d}(d\boldsymbol{x}) = 0$. Furthermore, the dimension of $\mathbb{Y}_n^{d+1}$ is $N_{n,d+1} := \frac{(2n+d-1)(n+d-2)!}{n!(d-1)!}$.

---

[4]Namely, $\mu_{\mathbb{S}^d}$ is the scaled version of the uniform probability measure $u_{\mathbb{S}^d}$ on $\mathbb{S}^d$ such that $u_{\mathbb{S}^d}(\cdot) = \mu_{\mathbb{S}^d}(\mathbb{S}^d)^{-1}\mu_{\mathbb{S}^d}(\cdot)$.

- **Addition Theorem** (e.g., Theorem 2.9 in (Atkinson & Han, 2012)). Let $(Y_{n,j})_{j \in [N_{n,d+1}]}$ be an orthonormal basis of $\mathbb{Y}_n^{d+1}$, i.e., $\forall j, k \in [N_{n,d+1}]$, $\int_{\mathbb{S}^d} Y_{n,j}(\boldsymbol{x}) Y_{n,k}(\boldsymbol{x}) \mu_{\mathbb{S}^d}(\mathrm{d}\boldsymbol{x}) = \mathbb{1}\{j = k\}$. Then, for any $\boldsymbol{x}, \widetilde{\boldsymbol{x}} \in \mathbb{S}^d$, the following equality holds:

$$\sum_{j=1}^{N_{n,d+1}} Y_{n,j}(\boldsymbol{x}) Y_{n,j}(\widetilde{\boldsymbol{x}}) = \frac{N_{n,d+1}}{|\mathbb{S}^d|} P_{n,d+1}(\boldsymbol{x}^\top \widetilde{\boldsymbol{x}}), \tag{7}$$

where $|\mathbb{S}^d| := 2\pi^{(d+1)/2}/\Gamma((d+1)/2)$ is the surface area of $\mathbb{S}^d$, and $P_{n,d+1} : [-1,1] \to \mathbb{R}$ is a Legendre polynomial defined as follows[5]:

$$P_{n,d+1}(t) = n! \Gamma\left(\frac{d}{2}\right) \sum_{k=0}^{\lfloor n/2 \rfloor} \frac{(-1)^k (1-t^2)^k t^{n-2k}}{4^k k! (n-2k)! \Gamma\left(k + \frac{d}{2}\right)}.$$

Specifically, $P_{n,2}(t) = \cos(n \arccos t)$ for $d = 1$.

In addition to the above properties, the most important known result for this paper is the fact that the spherical harmonics are eigenfunctions for a continuous kernel in $\mathbb{S}^d \times \mathbb{S}^d$, including the SE kernel. Below, we formally describe the Mercer decomposition of the SE kernel on the hypersphere.

**Lemma 2.9** (Theorem 2 in (Minh et al., 2006)). *Fix $d \in \mathbb{N}_+$ and $n \in \mathbb{N}$. Let $(Y_{n,j})_{j \in [N_{n,d+1}]}$ be an orthonormal basis of $\mathbb{Y}_n^{d+1}$ (i.e., $\forall j, k$, $\int_{\mathbb{S}^d} Y_{n,j}(\boldsymbol{x}) Y_{n,k}(\boldsymbol{x}) \mu_{\mathbb{S}^d}(\mathrm{d}\boldsymbol{x}) = \mathbb{1}\{j = k\}$). Then, $Y_{n,j}$ is an eigenfunction of the integral operator of the SE kernel on $\mathbb{S}^d$. Furthermore, the corresponding eigenvalue $\lambda_n$ satisfies*

$$\lambda_n \geq \underline{\lambda}_n := \left(\frac{2e}{\theta}\right)^n \frac{C_{d,\theta}}{(2n + d - 1)^{n + \frac{d}{2}}}, \tag{8}$$

*where $C_{d,\theta} > 0$ is the constant depending only on $d$ and $\theta$. Here, each eigenvalue $\lambda_n$ has multiplicity $N_{n,d+1}$.*

In addition, by combining Mercer's representation theorem with the above lemma, the RKHS norm of a function $\widetilde{f}$ of the form $\widetilde{f}(\boldsymbol{x}) := \sum_{n \in \mathbb{N}} \sum_{j=1}^{N_{n,d+1}} \alpha_{n,j} \sqrt{\lambda_n} Y_{n,j}(\boldsymbol{x})$ satisfies $\|\widetilde{f}\|_{\mathcal{H}_k(\mathbb{S}^d)} = \sqrt{\sum_{n \in \mathbb{N}} \sum_{j=1}^{N_{n,d+1}} \alpha_{n,j}^2}$, where, for any $n \in \mathbb{N}$, $(Y_{n,j})_{j \in [N_{n,d+1}]}$ is an orthonormal basis of $\mathbb{Y}_n^{d+1}$. We use this explicit representation of the RKHS norm in our proof.

## 3. Improved Lower Bounds for SE kernel on Hypersphere

The following theorems provide the formal statements of our improved lower bounds. The proofs are given in Ap-

---

[5]As described in (Atkinson & Han, 2012; Efthimiou & Frye, 2014), the term "Legendre polynomial" for $P_{n,d+1}(\cdot)$ is specific in spherical harmonics literature. In mathematical fields, $P_{n,3}(\cdot)$ is only referred to as Legendre polynomial.

pendix E.

**Theorem 3.1** (Improved simple regret lower bound for the SE kernel on the hypersphere). *Fix any $\delta \in (0, 1/3)$, $\epsilon \in (0, 1/2)$, $B > 0$, and $T \in \mathbb{N}_+$. Consider the GP bandit problem on $\mathcal{X} = \mathbb{S}^d$ with the SE kernel described in Section 2.1. Suppose that $d \in \mathbb{N}_+$ and $\theta > 0$ are fixed constants. Furthermore, suppose that there exists an algorithm that achieves $r_T \leq \epsilon$ with probability at least $1 - \delta$ for any $f \in \mathcal{H}_k(\mathcal{X}, B)$. Then, if $\epsilon/B$ is sufficiently small, it is necessary that*

$$T = \Omega\left(\frac{\sigma^2}{\epsilon^2}\left(\ln \frac{B}{\epsilon}\right)^d \left(\ln \ln \frac{B}{\epsilon}\right)^{-d} \ln \frac{1}{\delta}\right). \tag{9}$$

**Theorem 3.2** (Improved cumulative regret lower bound for the SE kernel on the hypersphere). *Consider the same setting as in Theorem 3.1. Furthermore, assume $\sigma^2 (\ln(1/\delta))/B^2 = O(T)$ with sufficiently small implied constant. Then, for any algorithm, there exists a function $f \in \mathcal{H}_k(\mathcal{X}, B)$ such that the following inequality holds with probability at least $\delta$:*

$$R_T = \Omega\left(\sqrt{T\sigma^2 \left(\ln \frac{B^2 T}{\sigma^2 \ln \frac{1}{\delta}}\right)^d \left(\ln \ln \frac{B^2 T}{\sigma^2 \ln \frac{1}{\delta}}\right)^{-d}}\right). \tag{10}$$

As described in (Cai & Scarlett, 2021), the above high-probability results also imply lower bounds on the expected regret. See Appendix B.

## 4. Proof Overview

Our technical contribution is the construction of a new class of hard functions. We first construct a new function class $\mathcal{F}_\epsilon^{\text{new}} \subset \mathcal{H}_k(\mathbb{S}^d, B)$ based on the spherical harmonics. After that, to improve the lower bound by following the proof strategy in (Scarlett et al., 2017), we replace Lemmas 2.1 and 2.4 with new counterparts tailored to $\mathcal{F}_\epsilon^{\text{new}}$.

### 4.1. Hard Function Construction on Hypersphere

The first step of our proof is to construct a hard function on $\mathbb{S}^d$. Intuitively, from Lemma 2.1 in the existing proof, we expect that the desired hard function should attain a large value around the small neighborhood of the maximizer, whereas the function values in the other regions are nearly 0. Based on this intuition, we consider approximating the Dirac delta function on $\mathbb{S}^d$ using spherical harmonics. Given a center point $\boldsymbol{z} \in \mathbb{S}^d$, let $\delta_{\boldsymbol{z}}(\boldsymbol{x})$ be the function that satisfies $\int_{\mathbb{S}^d} h(\boldsymbol{x}) \delta_{\boldsymbol{z}}(\boldsymbol{x}) \mu_{\mathbb{S}^d}(\mathrm{d}\boldsymbol{x}) = h(\boldsymbol{z})$ for any function $h$. As in the standard Euclidean space, for a function $h$, the best approximation $h_{\text{approx}}$ of $h$ under the basis $(Y_{n,j})_{j \in [N_{n,d+1}]}$ of $\mathbb{Y}_n^{d+1}$ is given by $h_{\text{approx}}(\boldsymbol{x}) = \sum_{j=1}^{N_{n,d+1}} \langle h, Y_{n,j} \rangle_{\mathbb{S}^d} Y_{n,j}(\boldsymbol{x})$ (Chapter 2.3 in Atkinson &

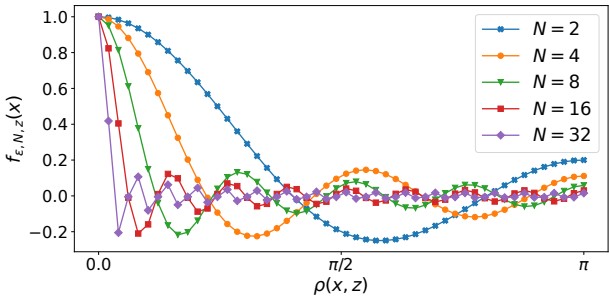

*Figure 2.* Visualization of the function $f_{\epsilon,N,z}$ with $\epsilon = 0.5$ and $d = 1$ (i.e., the function on the 2D circle $\mathbb{S}^1$). Note that the above plot takes the geodesic distance $\rho(x, z) := \arccos(x^\top z)$ from some center point $z \in \mathbb{S}^1$ as the horizontal axis. Intuitively, the number $N$ controls the sharpness of the behavior around the peak at $\rho(x, z) = 0$.

Han, 2012). Here, $\langle h_1, h_2 \rangle_{\mathbb{S}^d} := \int_{\mathbb{S}^d} h_1(x) h_2(x) \mu_{\mathbb{S}^d}(dx)$ is the inner product between functions $h_1$ and $h_2$ defined on $\mathbb{S}^d$. Then, for any $N$, we define the approximated delta function $b_{N,z}(x)$ centered at some $z \in \mathbb{S}^d$ as follows:

$$b_{N,z}(x) := \sum_{n=0}^{N} \sum_{j=1}^{N_{n,d+1}} \langle Y_{n,j}, \delta_z \rangle_{\mathbb{S}^d} Y_{n,j}(x) \tag{11}$$
$$= \sum_{n=0}^{N} \sum_{j=1}^{N_{n,d+1}} Y_{n,j}(z) Y_{n,j}(x),$$

where, for any $n$, $(Y_{n,j})_{j \in [N_{n,d+1}]}$ is an orthonormal basis of $\mathbb{Y}_n^{d+1}$. Based on $b_{N,z}$, for any $\epsilon > 0$, we also define the following scaled function $f_{\epsilon,N,z}$ of $b_{N,z}$ as follows:

$$f_{\epsilon,N,z}(x) = \frac{2\epsilon}{b_{N,z}(z)} b_{N,z}(x). \tag{12}$$

Figure 2 provides a visualization of $f_{\epsilon,N,z}$. By properly designing $N$ depending on $B$ and $\epsilon$, we can guarantee $\|f_{\epsilon,N,z}\|_{\mathcal{H}_k(\mathbb{S}^d)} \leq B$ and use it as an element $f_{z;\epsilon}^{\mathrm{new}}$ of our new hard function class $\mathcal{F}_\epsilon^{\mathrm{new}}$.

### 4.2. Properties of New Hard Function

The following lemma specifies the properties of $f_{z;\epsilon}^{\mathrm{new}}$ and serves as a counterpart to Lemma 2.1 in the original proof. The full proof is in Appendix D.

**Lemma 4.1** (Properties of new hard function). *Fix any $d \in \mathbb{N}_+$, $\epsilon > 0$, $B > 0$, $z \in \mathbb{S}^d$, and let $k : \mathbb{S}^d \times \mathbb{S}^d \to \mathbb{R}$ be the SE kernel with lengthscale parameter $\theta > 0$. Assume that $\epsilon/B$ is sufficiently small. Furthermore, define $f_{z;\epsilon}^{\mathrm{new}} : \mathbb{S}^d \to \mathbb{R}$ by $f_{z;\epsilon}^{\mathrm{new}} := f_{\epsilon,\bar{N},z}$, where $\bar{N} = \Theta\left(\left(\ln \frac{B}{\epsilon}\right)\left(\ln \ln \frac{B}{\epsilon}\right)^{-1}\right)$. In addition, set $w_\epsilon^{\mathrm{new}} := \Theta(\bar{N}^{-1})$. Then, the following three properties hold:*

1. *The function $f_{z;\epsilon}^{\mathrm{new}}$ attains its maximum at $z$. Furthermore, $f_{z;\epsilon}^{\mathrm{new}}(z) = 2\epsilon$ and $\forall x \in \mathbb{S}^d$, $|f_{z;\epsilon}^{\mathrm{new}}(x)| \leq 2\epsilon$*

*hold.*

2. *The $\epsilon$-optimal region satisfies $\{x \in \mathbb{S}^d \mid f_{z;\epsilon}^{\mathrm{new}}(z) - f_{z;\epsilon}^{\mathrm{new}}(x) \leq \epsilon\} \subset \{x \in \mathbb{S}^d \mid \rho(x, z) \leq w_\epsilon^{\mathrm{new}}/2 \text{ or } \rho(x, -z) \leq w_\epsilon^{\mathrm{new}}/2\}$, where $\rho(x, z) := \arccos(x^\top z)$ denotes the geodesic distance on $\mathbb{S}^d$.*

3. *The RKHS norm satisfies $\|f_{z;\epsilon}^{\mathrm{new}}\|_{\mathcal{H}_k(\mathbb{S}^d)} \leq B$.*

*Remark* 4.2 ($\epsilon$-optimal region guarantee in property 2). In the above lemma, property 2 suggests that the $\epsilon$-optimal region could be contained in neighborhoods around both $z$ and $-z$. We conjecture that this result is not intrinsic and could be improved in the future. However, even under this limitation, we can obtain the lower bound by defining a partition of $\mathbb{S}^d$ such that it includes both neighborhoods of $z$ and $-z$ (see Appendix C for details). Furthermore, note that this limitation only affects the constant factor in the final lower bounds.

Importantly, we can verify that the "width" $w_\epsilon^{\mathrm{new}} := \Theta\left(\left(\ln \frac{B}{\epsilon}\right)^{-1}\left(\ln \ln \frac{B}{\epsilon}\right)\right)$ around the function peak decreases faster than $w_\epsilon := \Theta(\ln^{-1/2}(B/\epsilon))$ in the existing proof. Intuitively, this suggests that our new function is more difficult to identify the peak location, leading to improved lower bounds.

**Proof Sketch for Properties 1 and 2.** For the proof, an important observation is that the function $b_{N,z}(x)$ can be transformed into a simple analytical expression via the addition theorem (Eq. (7)). Here, we focus on the case for $d = 1$, since the proof for $d = 1$ is a simplified version of that for general $d \geq 1$ and is suitable for explaining the core idea. From the addition theorem of spherical harmonics (Eq. (7)), we can write $b_{N,z}(x)$ as $b_{N,z}(x) = \sum_{n=0}^{N} \frac{N_{n,d+1}}{|\mathbb{S}^d|} P_{n,d+1}(x^\top z)$. Since we consider the case $d = 1$, $P_{n,d+1}(x^\top z) = \cos(n \arccos x^\top z)$, $|\mathbb{S}^d| = 2\pi$, $N_{0,d+1} = 1$, and $N_{n,d+1} = 2$ hold for $n \in \mathbb{N}_+$ from the definitions. Thus, $b_{N,z}(x)$ is simplified as

$$b_{N,z}(x) = \frac{1}{2\pi}\left[1 + 2\sum_{n=1}^{N} \cos(n \arccos x^\top z)\right]. \tag{13}$$

The term in parentheses of the above equation is known as the *Dirichlet kernel* (e.g., Chapter 15.2 in Bruckner et al., 1997). Then, from the above simple form, we can verify the properties 1 and 2 by elementary calculus. Finally, we note that the proof for general $d \geq 1$ is given by leveraging Gegenbauer polynomials (Szegö, 1939), instead of the Dirichlet kernel. See Appendix D.2 for details.

**Proof Sketch for Property 3.** From Mercer's representation theorem, the RKHS norm of $f_{\epsilon,N,z}$ is given by $\|f_{\epsilon,N,z}\|_{\mathcal{H}_k(\mathbb{S}^d)} =$

$\frac{2\epsilon}{b_{N,\boldsymbol{z}}(\boldsymbol{z})}\sqrt{\sum_{n=0}^{N}\sum_{j=1}^{N_{n,d+1}}(Y_{n,j}(\boldsymbol{z})/\sqrt{\lambda_n})^2}$, where $\lambda_n$ is the eigenvalue that satisfies Eq. (8). From this identity and the lower bound on $\lambda_n$ in Eq. (8), we find that the definition of $\bar{N}$ implies $\|f_{\epsilon,\bar{N},\boldsymbol{z}}\|_{\mathcal{H}_k(\mathbb{S}^d)} \leq B$ after elementary calculus.

**Construction of Finite Function Class.** By analogy of $\mathcal{F}_\epsilon$ (Definition 2.2) used in (Scarlett et al., 2017), we define the new function class $\mathcal{F}_\epsilon^{\text{new}} \subset \mathcal{H}_k(\mathbb{S}^d, B)$ based on Lemma 4.1 so that the $\epsilon$-optimal regions of the elements of $\mathcal{F}_\epsilon^{\text{new}}$ are distinct. Furthermore, from property 2 in Lemma 4.1, the new partition $(\mathcal{R}_{\boldsymbol{z};\epsilon}^{\text{new}})$ of $\mathbb{S}^d$ used in our proof is also defined. Due to space limitations, we leave the exact definitions to Definitions C.1 and C.2 in Appendix C.

### 4.3. Bounding Regret

The remaining parts of the proof bound the regret using $\mathcal{F}_\epsilon^{\text{new}}$. To obtain final results via the existing proof strategy summarized in Section 2.2.2, we leverage the following lemma, which serves as a replacement of Lemma 2.4.

**Lemma 4.3.** *Let $\mathcal{F}_\epsilon^{\text{new}}$, $\mathcal{N}_\epsilon^{\text{new}}$, and $(\mathcal{R}_{\boldsymbol{z};\epsilon}^{\text{new}})_{\boldsymbol{z}\in\mathcal{N}_\epsilon^{\text{new}}}$ be the function class, the finite index set, and the partition defined in Definitions C.1 and C.2. Then, we have $\forall \boldsymbol{z} \in \mathcal{N}_\epsilon^{\text{new}}$, $\sum_{\widetilde{\boldsymbol{z}}\in\mathcal{N}_\epsilon^{\text{new}}} \sup_{\boldsymbol{x}\in\mathcal{R}_{\widetilde{\boldsymbol{z}};\epsilon}^{\text{new}}} \left(f_{\widetilde{\boldsymbol{z}};\epsilon}^{\text{new}}(\boldsymbol{x})\right)^2 \leq \widehat{C}_d \epsilon^2$, for some constant $\widehat{C}_d > 0$ depending only on $d$.*

As with the proof of Lemma 4.1, the core elements of the proof of Lemma 4.3 are the simplifications via the addition theorem of spherical harmonics. The full proof is provided in Appendix D.4.

Finally, by following the existing proof strategy in (Cai & Scarlett, 2021) and using the aforementioned lemmas, we obtain the desired lower bounds.

## 5. Improved Information Gain Upper Bound

One intriguing feature of our lower bounds is the presence of an additional term of the form $\Omega((\ln\ln T)^{-d/2})$. By examining our proof, we observe that this term arises from the logarithmic factor in the exponent of the eigenvalue $\lambda_n := \Omega(\exp(-n\ln n))$ (see Eq. (8)). We leave the rigorous examination of whether the $\Omega((\ln\ln T)^{-d/2})$-term is essential the future work. On the other hand, at present, we conjecture that $\Omega((\ln\ln T)^{-d/2})$ term is unavoidable. The reason for this conjecture is that the $O((\ln\ln T)^{-d/2})$ term also naturally appears in the upper bound on the regret through the MIG $\gamma_T$. We show this as follows.

**Theorem 5.1** (Improved upper bound of MIG). *Fix any $d \in \mathbb{N}_+$, $\lambda > 0$, and $T \in \mathbb{N}_+$. Let $\mathcal{X}$ be either a $d$-dimensional compact input domain in $\mathbb{R}^d$ or a hyperspherical input domain $\mathcal{X} \subset \mathbb{S}^d$. Let $\gamma_T$ denote the MIG of the SE*

*kernel on $\mathcal{X}$, which is defined as follows:*

$$\gamma_T = \frac{1}{2} \sup_{\boldsymbol{x}_1,\ldots,\boldsymbol{x}_T\in\mathcal{X}} \ln\det(\boldsymbol{I}_T + \lambda^{-2}\mathbf{K}_T), \quad (14)$$

*where $\mathbf{K}_T := [k(\boldsymbol{x}_i,\boldsymbol{x}_j)]_{i,j\in[T]} \in \mathbb{R}^{T\times T}$ is the kernel matrix. Furthermore, $k$ is the SE kernel defined in Eq. (1). Then, $\gamma_T = O((\ln T)^{d+1}(\ln\ln T)^{-d})$.*

Note that the above result is applicable to a general compact input domain in $\mathbb{R}^d$ beyond $\mathbb{S}^d$. The basic proof strategy of the above theorem follows that of the existing MIG upper bound in (Iwazaki, 2025b). The difference in our proof is a more precise treatment of the rate of decay of the kernel eigenvalues in the existing proof. See the full proof in Appendix F. Since the above result improves the existing $O((\ln T)^{d+1})$ upper bound on MIG (Iwazaki, 2025b; Srinivas et al., 2010), this is of independent interest. For example, as described in Section 1.1, the cumulative regret upper bounds of many existing algorithms are quantified as $O^*(\sqrt{T\gamma_T})$ or $O^*(\gamma_T\sqrt{T})$; thus, Theorem 5.1 immediately provides $O(\sqrt{(\ln T)^d})$ or $O((\ln\ln T)^d)$ improvements for such algorithms.

## 6. Discussion

One desired future direction is to extend our analysis to a compact input domain $\mathcal{X}$ on $\mathbb{R}^d$. For simplicity, we assume $\mathcal{X} = [0,1]^d$ in this section. A natural direction is to extend our function construction in Section 4.1 by using some eigenfunctions on $\mathbb{R}^d$. For example, under the Gaussian measure, it is known that the corresponding eigenfunctions $(\phi_n^{\text{Gauss}})_{n\in\mathbb{N}}$ and eigenvalues $(\lambda_n^{\text{Gauss}})_{n\in\mathbb{N}}$ of the SE kernel are given in explicit form (e.g., Chapter 4.3.1 in Rasmussen & Williams, 2005). Using these results and following the same philosophy as in Section 4.1, we can adapt our proof strategy by redefining $b_{N,\boldsymbol{z}}$ (Eq. (11)) as $b_{N,\boldsymbol{z}}(\cdot) = \sum_{n=0}^{N} \phi_n^{\text{Gauss}}(\boldsymbol{z})\phi_n^{\text{Gauss}}(\cdot)$. However, in our attempts, the logarithmic factors in the resulting lower bound scale as $\Omega(\sqrt{(\ln T)^{d/2}})$, which exhibits no improvement. We provide the details in Appendix G. We conjecture that this is because the Gaussian measure has unbounded support, whereas the problem is defined on a bounded domain. Therefore, we believe that a promising future direction is to leverage the eigensystem $(\phi_n, \lambda_n)$ associated with a measure whose support matches the input domain $\mathcal{X} := [0,1]^d$. For example, the uniform measure on $[0,1]^d$ and the corresponding eigensystem $(\phi_n^{\text{unif}}, \lambda_n^{\text{unif}})$ of the SE kernel may be a natural choice. We leave a further examination of this direction for future work.

## 7. Conclusion

We provide the improved lower bounds for the SE kernel on the hypersphere. Our results fill the gap in the dimension-

dependent logarithmic factors between the existing lower and upper bounds. We also discuss promising directions for extending our proof idea to a general compact input domain. Although the current analysis is limited to the hypersphere, we believe that our results and core proof techniques represent an important milestone toward improving the lower bounds for the SE kernel on general compact domains.

## Impact Statement

This paper presents work whose goal is to advance the field of Machine Learning. There are many potential societal consequences of our work, none which we feel must be specifically highlighted here.

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

# A. Proofs of Corollaries 2.5 and 2.6

To obtain the lower bound on the hypersphere, we first modify the definition of the finite function class.

**Definition A.1** (Finite function class construction based on $g_\epsilon$ on the hypersphere). Define $M_\epsilon^{(s)}$ as $M_\epsilon^{(s)} = P(\mathbb{S}^d, w_\epsilon, \|\cdot\|_\infty)$, where $P(\mathbb{S}^d, w_\epsilon, \|\cdot\|_\infty)$ is the $w_\epsilon$-packing number of $\mathbb{S}^d$ with respect to the infinity-norm $\|\cdot\|_\infty$. Furthermore, let $\mathcal{N}_\epsilon^{(s)} \subset \mathbb{S}^d$ be a $w_\epsilon$-separated set of $\mathbb{S}^d$ such that $|\mathcal{N}_\epsilon^{(s)}| = M_\epsilon^{(s)}$. Then, we define the function class $\mathcal{F}_\epsilon^{(s)} := (f_{\boldsymbol{z};\epsilon}^{(s)})_{\boldsymbol{z} \in \mathcal{N}_\epsilon^{(s)}}$, where $f_{\boldsymbol{z};\epsilon}^{(s)} : \mathbb{S}^d \to [-2\epsilon, 2\epsilon]$ is defined as $f_{\boldsymbol{z};\epsilon}^{(s)}(\boldsymbol{x}) = g_\epsilon(\boldsymbol{x} - \boldsymbol{z})$. Here, $g_\epsilon$ is the function defined in Lemma 2.1 with the dimension $d+1$.

In addition, we also define the partition of $\mathbb{S}^d$ as follows.

**Definition A.2** (Partition of hypersphere). Let $\mathcal{N}_\epsilon^{(s)}$ be the $w_\epsilon$-separated set of $\mathbb{S}^d$ defined in Lemma A.1. For any $\boldsymbol{z} \in \mathcal{N}_\epsilon^{(s)}$, let us define the partition $(\mathcal{R}_{\boldsymbol{z};\epsilon}^{(s)})_{\boldsymbol{z} \in \mathcal{N}_\epsilon^{(s)}}$ of $\mathbb{S}^d$ as the Voronoi decomposition regarding the infinity-norm $\|\cdot\|_\infty$. Namely, $(\mathcal{R}_{\boldsymbol{z};\epsilon}^{(s)})_{\boldsymbol{z} \in \mathcal{N}_\epsilon^{(s)}}$ satisfies $\boldsymbol{z} \in \mathrm{argmin}_{\widetilde{\boldsymbol{z}} \in \mathcal{N}_\epsilon^{(s)}} \|\widetilde{\boldsymbol{z}} - \boldsymbol{x}\|_\infty$ for any $\boldsymbol{x} \in \mathcal{R}_{\boldsymbol{z};\epsilon}^{(s)}$.

Here, note the following facts:

- From the definition of $w_\epsilon$-separated set, any two distinct regions in $(\mathcal{R}_{\boldsymbol{z};\epsilon}^{(s)})$ do not share an $\epsilon$-optimal region on $\mathbb{S}^d$. Furthermore, $2\epsilon = f_{\boldsymbol{z};\epsilon}^{(s)}(\boldsymbol{z}) = \max_{\boldsymbol{x} \in \mathbb{S}^d} f_{\boldsymbol{z};\epsilon}^{(s)}(\boldsymbol{x})$.

- As with the case where the input domain is $\mathcal{X} = [0,1]^d$, the RKHS norm of $f_{\boldsymbol{z};\epsilon}^{(s)}$ is less than or equal to $B$, since the input shift and restriction do not increase the RKHS norm.

- Since $P(\mathbb{S}^d, w_\epsilon, \|\cdot\|_2) = \Theta(w_\epsilon^{-d})$ (e.g., Corollary 4.2.13 in Vershynin, 2018), the packing number $P(\mathbb{S}^d, w_\epsilon, \|\cdot\|_\infty)$ also satisfies $P(\mathbb{S}^d, w_\epsilon, \|\cdot\|_\infty) = \Theta(w_\epsilon^{-d})$ from the norm equivalence of $\|\cdot\|_\infty$ and $\|\cdot\|_2$.

By noting the above properties and applying the same proof in (Cai & Scarlett, 2021), we can obtain the desired statements. Here, the remaining thing we need to show is the following lemma, which replaces Lemma 2.4.

**Lemma A.3** (Replacement of Lemma 2.4). *Let $\mathcal{F}_\epsilon^{(s)}$, $\mathcal{N}_\epsilon^{(s)}$, and $(\mathcal{R}_{\boldsymbol{z};\epsilon}^{(s)})_{\boldsymbol{z} \in \mathcal{N}_\epsilon^{(s)}}$ be the function class, the finite index set, and the decomposition of $\mathbb{S}^d$ in Definitions A.1 and A.2. Then, we have*

$$\forall \boldsymbol{z} \in \mathcal{N}_\epsilon^{(s)}, \quad \sum_{\widetilde{\boldsymbol{z}} \in \mathcal{N}_\epsilon^{(s)}} \sup_{\boldsymbol{x} \in \mathcal{R}_{\boldsymbol{z};\epsilon}^{(s)}} \left( f_{\widetilde{\boldsymbol{z}};\epsilon}^{(s)}(\boldsymbol{x}) \right)^2 \le C_d \epsilon^2, \tag{15}$$

*where $C_d > 0$ is a constant depending on $d$.*

*Proof.* Fix any $\boldsymbol{z} \in \mathcal{N}_\epsilon^{(s)}$ and $\epsilon > 0$. Here, we have

$$\sum_{\widetilde{\boldsymbol{z}} \in \mathcal{N}_\epsilon^{(s)}} \sup_{\boldsymbol{x} \in \mathcal{R}_{\boldsymbol{z};\epsilon}^{(s)}} \left( f_{\widetilde{\boldsymbol{z}};\epsilon}^{(s)}(\boldsymbol{x}) \right)^2 = \frac{4\epsilon^2}{h^2(\boldsymbol{0})} \sum_{\widetilde{\boldsymbol{z}} \in \mathcal{N}_\epsilon^{(s)}} \sup_{\boldsymbol{x} \in \mathcal{R}_{\boldsymbol{z};\epsilon}^{(s)}} h^2 \left( \frac{\zeta(\boldsymbol{x} - \widetilde{\boldsymbol{z}})}{w_\epsilon} \right). \tag{16}$$

Then, we decompose the summation based on the distance from $\mathcal{R}_{\boldsymbol{z};\epsilon}^{(s)}$ as follows:

$$\sum_{\widetilde{\boldsymbol{z}} \in \mathcal{N}_\epsilon^{(s)}} \sup_{\boldsymbol{x} \in \mathcal{R}_{\boldsymbol{z};\epsilon}^{(s)}} h^2 \left( \frac{\zeta(\boldsymbol{x} - \widetilde{\boldsymbol{z}})}{w_\epsilon} \right) = \sup_{\boldsymbol{x} \in \mathcal{R}_{\boldsymbol{z};\epsilon}^{(s)}} h^2 \left( \frac{\zeta(\boldsymbol{x} - \boldsymbol{z})}{w_\epsilon} \right) + \sum_{i \in \mathbb{N}} \sum_{\widetilde{\boldsymbol{z}} \in \mathcal{Z}_i^{(s)}} \sup_{\boldsymbol{x} \in \mathcal{R}_{\boldsymbol{z};\epsilon}^{(s)}} h^2 \left( \frac{\zeta(\boldsymbol{x} - \widetilde{\boldsymbol{z}})}{w_\epsilon} \right) \tag{17}$$

$$\le h^2(\boldsymbol{0}) + \sum_{i \in \mathbb{N}} |\mathcal{Z}_i^{(s)}| \sup_{\widetilde{\boldsymbol{z}} \in \mathcal{Z}_i^{(s)}} \sup_{\boldsymbol{x} \in \mathcal{R}_{\boldsymbol{z};\epsilon}^{(s)}} h^2 \left( \frac{\zeta(\boldsymbol{x} - \widetilde{\boldsymbol{z}})}{w_\epsilon} \right), \tag{18}$$

where $\mathcal{Z}_i^{(s)} = \{\widetilde{\boldsymbol{z}} \in \mathcal{N}_\epsilon^{(s)} \mid iw_\epsilon/2 < \inf_{\boldsymbol{x} \in \mathcal{R}_{\boldsymbol{z};\epsilon}^{(s)}} \|\boldsymbol{x} - \widetilde{\boldsymbol{z}}\|_2 \leq (i+1)w_\epsilon/2\}$. Here, from Lemma H.6, we have

$$\sum_{\widetilde{\boldsymbol{z}} \in \mathcal{N}_\epsilon^{(s)}} \sup_{\boldsymbol{x} \in \mathcal{R}_{\boldsymbol{z};\epsilon}^{(s)}} h^2\left(\frac{\zeta(\boldsymbol{x} - \widetilde{\boldsymbol{z}})}{w_\epsilon}\right) \leq h^2(\boldsymbol{0}) + \widetilde{C}_d \left[\sup_{\widetilde{\boldsymbol{z}} \in \mathcal{Z}_0^{(s)}} \sup_{\boldsymbol{x} \in \mathcal{R}_{\boldsymbol{z};\epsilon}^{(s)}} h^2\left(\frac{\zeta(\boldsymbol{x} - \widetilde{\boldsymbol{z}})}{w_\epsilon}\right) + \sum_{i \in \mathbb{N}_+} i^{d+1} \sup_{\widetilde{\boldsymbol{z}} \in \mathcal{Z}_i^{(s)}} \sup_{\boldsymbol{x} \in \mathcal{R}_{\boldsymbol{z};\epsilon}^{(s)}} h^2\left(\frac{\zeta(\boldsymbol{x} - \widetilde{\boldsymbol{z}})}{w_\epsilon}\right)\right] \tag{19}$$

$$\leq h^2(\boldsymbol{0}) + \widetilde{C}_d \left[h^2(\boldsymbol{0}) + \sum_{i \in \mathbb{N}_+} i^{d+1} \sup_{\widetilde{\boldsymbol{z}} \in \mathcal{Z}_i^{(s)}} \sup_{\boldsymbol{x} \in \mathcal{R}_{\boldsymbol{z};\epsilon}^{(s)}} h^2\left(\frac{\zeta(\boldsymbol{x} - \widetilde{\boldsymbol{z}})}{w_\epsilon}\right)\right], \tag{20}$$

where $\widetilde{C}_d > 0$ is some constant depending only on $d$. Furthermore, as described in (Scarlett et al., 2017), $h(\cdot)$ decreases faster than any finite power of $1/\|\boldsymbol{x}\|_2$ as $\|\boldsymbol{x}\|_2 \to \infty$. This implies that, for any $c > 0$ and $d \in \mathbb{N}_+$, there exists constant a $C_{c,d} > 0$ such that $\forall \boldsymbol{x} \in \{\widetilde{\boldsymbol{x}} \in \mathbb{R}^{d+1} \mid \|\widetilde{\boldsymbol{x}}\|_2 \geq c\}$, $|h(\boldsymbol{x})| \leq C_{c,d}/\|\boldsymbol{x}\|_2^{(d+3)/2}$. We set $c > 0$ as $c = \zeta/2$. Then, since $\frac{\|\boldsymbol{x} - \widetilde{\boldsymbol{z}}\|_2 \zeta}{w_\epsilon} > \frac{(w_\epsilon/2)\zeta}{w_\epsilon} = \zeta/2$ for any $\widetilde{\boldsymbol{z}} \in \mathcal{Z}_i^{(s)}$ and $\boldsymbol{x} \in \mathcal{R}_{\boldsymbol{z};\epsilon}^{(s)}$ with $i \geq 1$, we have the following inequalities for some constant $\widehat{C}_d > 0$:

$$\sum_{i \in \mathbb{N}_+} i^{d+1} \sup_{\widetilde{\boldsymbol{z}} \in \mathcal{Z}_i^{(s)}} \sup_{\boldsymbol{x} \in \mathcal{R}_{\boldsymbol{z};\epsilon}^{(s)}} h^2\left(\frac{\zeta(\boldsymbol{x} - \widetilde{\boldsymbol{z}})}{w_\epsilon}\right) \leq \widehat{C}_d \sum_{i \in \mathbb{N}_+} i^{d+1} \sup_{\widetilde{\boldsymbol{z}} \in \mathcal{Z}_i^{(s)}} \sup_{\boldsymbol{x} \in \mathcal{R}_{\boldsymbol{z};\epsilon}^{(s)}} \left(\frac{1}{(\|\boldsymbol{x} - \widetilde{\boldsymbol{z}}\|_2 \zeta/w_\epsilon)^{(d+3)/2}}\right)^2 \tag{21}$$

$$= \widehat{C}_d \sum_{i \in \mathbb{N}_+} i^{d+1} \sup_{\widetilde{\boldsymbol{z}} \in \mathcal{Z}_i^{(s)}} \sup_{\boldsymbol{x} \in \mathcal{R}_{\boldsymbol{z};\epsilon}^{(s)}} \left(\frac{w_\epsilon}{\|\boldsymbol{x} - \widetilde{\boldsymbol{z}}\|_2 \zeta}\right)^{d+3} \tag{22}$$

$$\leq \widehat{C}_d \left(\frac{2}{\zeta}\right)^{d+3} \sum_{i \in \mathbb{N}_+} i^{-2} \tag{23}$$

$$= \frac{\pi^2 \widehat{C}_d}{6} \left(\frac{2}{\zeta}\right)^{d+3}, \tag{24}$$

where the second inequality follows from the definition of $\mathcal{Z}_i^{(s)}$. By combining the above inequality with Eqs. (16) and (20), we have

$$\sum_{\widetilde{\boldsymbol{z}} \in \mathcal{N}_\epsilon^{(s)}} \sup_{\boldsymbol{x} \in \mathcal{R}_{\boldsymbol{z};\epsilon}^{(s)}} \left(f_{\widetilde{\boldsymbol{z}};\epsilon}^{(s)}(\boldsymbol{x})\right)^2 \leq \epsilon^2 \left[\frac{4}{h^2(\boldsymbol{0})}\left(h^2(\boldsymbol{0}) + \widetilde{C}_d h^2(\boldsymbol{0}) + \widetilde{C}_d \frac{\pi^2 \widehat{C}_d}{6}\left(\frac{2}{\zeta}\right)^{d+3}\right)\right]. \tag{25}$$

Note that the term in the square brackets in the above inequality only depends on $d$; thus, we obtain the desired statement. $\quad\square$

## B. Lower Bounds for Expected Regret

In this section, we present the following corollaries, which provide lower bounds on the expected regret.

**Corollary B.1** (Expected simple regret lower bound for the SE kernel on the hypersphere). *Fix any $\epsilon \in (0, 1/2)$, $B > 0$, and $T \in \mathbb{N}_+$. Let us consider the GP bandit problem on $\mathcal{X} = \mathbb{S}^d$ with the SE kernel described in Section 2.1. Suppose that $d \in \mathbb{N}_+$ and $\theta > 0$ are fixed constants. Furthermore, suppose that there exists an algorithm that achieves $\mathbb{E}[r_T] \leq \epsilon$ for any $f \in \mathcal{H}_k(\mathcal{X}, B)$. Then, if $\epsilon/B$ is sufficiently small, it is necessary that*

$$T = \Omega\left(\frac{\sigma^2}{\epsilon^2}\left(\ln\frac{B}{\epsilon}\right)^d \left(\ln\ln\frac{B}{\epsilon}\right)^{-d}\right). \tag{26}$$

*Proof.* Fix any algorithm and set $\delta = 1/6$. If $\mathbb{E}[r_T] \leq \epsilon$, the probability $\mathbb{P}(r_T > 6\epsilon)$ must be less than $\delta$; otherwise, $\mathbb{E}[r_T] \geq \mathbb{E}[\mathbb{1}\{r_T > 6\epsilon\}r_T] > \mathbb{P}(r_T \geq 6\epsilon)6\epsilon = \epsilon$, which is a contradiction. Thus, if $\mathbb{E}[r_T] \leq \epsilon$ holds for any $f \in \mathcal{H}_k(\mathcal{X}, B)$, then, $r_T \leq 6\epsilon$ holds with probability at least $1 - \delta$ for any $f \in \mathcal{H}_k(\mathcal{X}, B)$. By applying Theorem 3.1 with $6\epsilon$ and adjusting the constant factor, we obtain the desired statement. $\quad\square$

**Corollary B.2** (Expected cumulative regret lower bound for SE kernel in hypersphere). *Let us consider the GP bandit problem on $\mathcal{X} = \mathbb{S}^d$ with the SE kernel described in Section 2.1. Suppose that $d \in \mathbb{N}_+$ and $\theta > 0$ are fixed constants, and $B$,*

$\sigma^2$, $\delta > 0$, and $T \in \mathbb{N}_+$ satisfy $\sigma^2/B^2 = O(T)$ with sufficiently small implied constant. Then, for any algorithm, there exists a function $f \in \mathcal{H}_k(\mathcal{X}, B)$ such that the following inequality holds:

$$\mathbb{E}[R_T] = \Omega\left(\sqrt{T\sigma^2 \left(\ln \frac{B^2 T}{\sigma^2}\right)^d \left(\ln \ln \frac{B^2 T}{\sigma^2}\right)^{-d}}\right).$$

*Proof.* We immediately obtain the desired statement by setting a constant $\delta \in (0, 1/3)$ (e.g., $\delta = 1/6$) in Theorem 3.2. $\square$

## C. Definition of New Function Class and Partition of Hypersphere

In this section, we define our new function class and partition of $\mathbb{S}^d$. As described in Remark 4.2, we define the function class and the partition so that the $w_\epsilon^{\text{new}}/2$-neighborhoods of $z$ and $-z$ do not overlap, as follows.

**Definition C.1** (New function class). Let $f_{z;\epsilon}^{\text{new}}$ and $w_\epsilon^{\text{new}}$ be the function and parameter defined in Lemma 4.1, respectively. Define $\mathbb{S}_+^d(\cdot)$ as $\mathbb{S}_+^d(w_\epsilon^{\text{new}}) = \{\widetilde{x} \in \mathbb{S}^d \mid \rho(\widetilde{x}, \mathbb{S}_{\text{equator}}^d) > w_\epsilon^{\text{new}}/2 \text{ and } \widetilde{x}_{d+1} > 0\}$, where $\rho(\widetilde{x}, \mathbb{S}_{\text{equator}}^d) = \inf_{x \in \mathbb{S}_{\text{equator}}^d} \rho(\widetilde{x}, x)$ and $\mathbb{S}_{\text{equator}}^d = \{x \in \mathbb{S}^d \mid x_{d+1} = 0\}$. Define $M_\epsilon^{\text{new}}$ as $M_\epsilon^{\text{new}} = P(w_\epsilon^{\text{new}}, \mathbb{S}_+^d(w_\epsilon^{\text{new}}), \rho)$, where $P(w_\epsilon^{\text{new}}, \mathbb{S}_+^d(w_\epsilon^{\text{new}}), \rho)$ is the $w_\epsilon^{\text{new}}$-packing number of $\mathbb{S}_+^d(w_\epsilon^{\text{new}})$ regarding the geodesic distance $\rho$. Furthermore, let $\mathcal{N}_\epsilon^{\text{new}} \subset \mathcal{X}$ be a $w_\epsilon^{\text{new}}$-separated set of $\mathbb{S}_+^d(w_\epsilon^{\text{new}})$ such that $|\mathcal{N}_\epsilon^{\text{new}}| = M_\epsilon^{\text{new}}$. Then, we define the function class $\mathcal{F}_\epsilon^{\text{new}}$ as $\mathcal{F}_\epsilon^{\text{new}} = (f_{z;\epsilon}^{\text{new}})_{z \in \mathcal{N}_\epsilon^{\text{new}}}$.

**Definition C.2** (New Partition of $\mathbb{S}^d$). Let $\mathcal{N}_\epsilon^{\text{new}}$ be the $w_\epsilon^{\text{new}}$-separated set of $\mathbb{S}_+^d(w_\epsilon^{\text{new}})$ defined in Definition C.1. For any $z \in \mathcal{N}_\epsilon^{\text{new}}$, let us define the partition $(\mathcal{R}_z^+)_{z \in \mathcal{N}_\epsilon^{\text{new}}}$ of $\mathbb{S}_+^d := \{x \in \mathbb{S}^d \mid x_{d+1} \geq 0\}$ as the Voronoi decomposition regarding the geodesic distance $\rho$. Namely, $(\mathcal{R}_z^+)_{z \in \mathcal{N}_\epsilon^{\text{new}}}$ satisfies $z \in \arg\min_{\widetilde{z} \in \mathcal{N}_\epsilon^{\text{new}}} \rho(\widetilde{z}, x)$ for any $x \in \mathcal{R}_z^+$. Furthermore, we define $\mathcal{R}_z^-$ as $\mathcal{R}_z^- := \{-x \mid x \in \mathcal{R}_z^+\} \setminus \mathbb{S}_{\text{equator}}^d$. Then, we define the partition $(\mathcal{R}_{z;\epsilon}^{\text{new}})_{z \in \mathcal{N}_\epsilon^{\text{new}}}$ of $\mathbb{S}^d$ as $\mathcal{R}_{z;\epsilon}^{\text{new}} = \mathcal{R}_z^+ \cup \mathcal{R}_z^-$.

From the above construction, we can observe the following facts.

- For any $z \in \mathcal{N}_\epsilon^{\text{new}}$, the region $\mathcal{R}_{z;\epsilon}^{\text{new}}$ contains two $w_\epsilon^{\text{new}}/2$-balls centered at $z$ and $-z$. This implies that, for any $z \in \mathcal{N}_\epsilon^{\text{new}}$, the region $\mathcal{R}_{z;\epsilon}^{\text{new}}$ contains the $\epsilon$-optimal region of $f_{z;\epsilon}^{\text{new}}$, while the region $\mathcal{R}_{z;\epsilon}^{\text{new}}$ does not include any $\epsilon$-optimal point of $f_{\widetilde{z};\epsilon}^{\text{new}}$ for $\widetilde{z} \neq z$.

- Provided that $\epsilon/B$ is sufficiently small, $w_\epsilon^{\text{new}}$ is also sufficiently small. Thus, for sufficiently small $\epsilon/B$, we have $P(\mathbb{S}_+^d(w_{d,\theta}), w_\epsilon^{\text{new}}, \|\cdot\|_2) \leq P(\mathbb{S}_+^d(w_\epsilon^{\text{new}}), w_\epsilon^{\text{new}}, \|\cdot\|_2) \leq P(\mathbb{S}_+^d, w_\epsilon^{\text{new}}, \|\cdot\|_2)$ for some small constant $w_{d,\theta} > 0$. Since $P(\mathbb{S}_+^d, w_\epsilon^{\text{new}}, \|\cdot\|_2) = \Theta((w_\epsilon^{\text{new}})^{-d})$ and $P(\mathbb{S}_+^d(w_{d,\theta}), w_\epsilon^{\text{new}}, \|\cdot\|_2) = \Theta((w_\epsilon^{\text{new}})^{-d})$, we have $P(\mathbb{S}_+^d(w_\epsilon^{\text{new}}), w_\epsilon^{\text{new}}, \|\cdot\|_2) = \Theta((w_\epsilon^{\text{new}})^{-d})$. By noting the relation $\forall z, \widetilde{z} \in \mathbb{S}^d, \frac{2}{\pi}\rho(z, \widetilde{z}) \leq \|z - \widetilde{z}\|_2 \leq \rho(z, \widetilde{z})$, we also have $M_\epsilon^{\text{new}} = P(\mathbb{S}_+^d(w_\epsilon^{\text{new}}), w_\epsilon^{\text{new}}, \rho) = \Theta((w_\epsilon^{\text{new}})^{-d})$ for sufficiently small $\epsilon/B$.

From the first property, we can apply Lemma 2.3 using the event $\mathcal{A}$ defined in the proof in (Cai & Scarlett, 2021) (see the proofs in Appendix E for details). Furthermore, from the second property, we can observe that $M_\epsilon^{\text{new}}$ increases with the same order as $M_\epsilon$ in the original proof.

## D. Proofs of Lemmas in Section 4

### D.1. Proof of Properties 1 and 2 of Lemma 4.1 for $d = 1$

*Proof.* As described in Section 4.2, when $d = 1$, the function $b_{N,z}$ is written as follows:

$$b_{N,z}(x) = \frac{1}{2\pi}\left[1 + 2\sum_{n=1}^N \cos(n \arccos x^\top z)\right]. \tag{27}$$

**Regarding Property 1.** Since $-N \leq \sum_{n=1}^N \cos(n \arccos x^\top z) \leq N$, $\forall x \in \mathbb{S}^d$, $|b_{N,z}(x)| \leq \frac{1+2N}{2\pi}$. Furthermore, $b_{N,z}(z) = \frac{1+2N}{2\pi}$, which implies that $z$ is a maximizer of $b_{N,z}$. Therefore, the center point $z$ is also a maximizer of $f_{\epsilon,N,z}(x)$, and $f_{\epsilon,N,z}(z) = 2\epsilon$ holds. Furthermore,

$$|f_{\epsilon,N,z}(x)| = 2\epsilon \frac{|b_{N,z}(x)|}{|b_{N,z}(z)|} \leq 2\epsilon. \tag{28}$$

**Regarding Property 2.** Let us define $\widetilde{b}_N : [0, \pi] \to \mathbb{R}$ as $\widetilde{b}_N(t) = 1 + 2\sum_{n=1}^{N}\cos(nt)$. From the definition of $f_{\epsilon,N,\boldsymbol{z}}$ and $b_{N,\boldsymbol{z}}$, it is enough to show that $\widetilde{b}_N(t) \leq \widetilde{b}_N(0)/2$ holds for all $t \in [\pi/N, \pi]$ and $N \geq 1$ (i.e., the implied constant of $w_{\epsilon}^{\mathrm{new}}$ can be taken as $2\pi$). Here, for $t > 0$, the following identity of the Dirichlet kernel is known (Bruckner et al., 1997):

$$\widetilde{b}_N(t) = \frac{\sin\left(\left(N + \frac{1}{2}\right)t\right)}{\sin\left(\frac{t}{2}\right)}. \tag{29}$$

Thus, for any $t \in [\pi/N, \pi]$, we obtain the desired inequality as follows:

$$\widetilde{b}_N(t)\widetilde{b}_N(0)^{-1} = \frac{\sin\left(\left(N + \frac{1}{2}\right)t\right)}{(1 + 2N)\sin\left(\frac{t}{2}\right)} \tag{30}$$

$$\leq \frac{1}{(1 + 2N)\sin\left(\frac{\pi}{2N}\right)} \tag{31}$$

$$\leq \frac{1}{(1 + 2N)\frac{\pi}{2N}\frac{2}{\pi}} \tag{32}$$

$$\leq \frac{1}{2}, \tag{33}$$

where the first inequality follows from $\sin\left(\left(N + \frac{1}{2}\right)t\right) \leq 1$ and $\sin\left(\frac{t}{2}\right) \geq \sin\left(\frac{\pi}{2N}\right) > 0$, and the second inequality follows from the inequality: $\forall x \in [0, \pi/2], \sin(x) \geq \frac{2}{\pi}x$. $\qquad\square$

*Remark* D.1. As shown in the above proof, if we assume $d = 1$, we can show that the $\epsilon$-optimal region is completely contained in the neighborhood of $\boldsymbol{z}$. This is not the case in the general proof for $d \geq 1$ in the next subsection.

### D.2. Proof of Properties 1 and 2 of Lemma 4.1 for general $d \geq 1$

*Proof.* As with the proof for $d = 1$, we first rewrite $b_{N,\boldsymbol{z}}$ using the addition theorem as follows:

$$b_{N,\boldsymbol{z}}(\boldsymbol{x}) = \sum_{n=0}^{N}\frac{N_{n,d+1}}{|\mathbb{S}^d|}P_{n,d+1}(\boldsymbol{x}^\top\boldsymbol{z}). \tag{34}$$

For $d \geq 2$, Legendre polynomial $P_{n,d+1}$ is written in terms of the Gegenbauer polynomial $C_n^{(\eta_d)}$ as follows (Lemma H.2):

$$P_{n,d+1}(t) = \frac{n!(d-2)!}{(n+d-2)!}C_n^{(\eta_d)}(t), \tag{35}$$

where $\eta_d = (d-1)/2$. Therefore, we have

$$b_{N,\boldsymbol{z}}(\boldsymbol{x}) = \frac{1}{|\mathbb{S}^d|}\sum_{n=0}^{N}N_{n,d+1}\frac{n!(d-2)!}{(n+d-2)!}C_n^{(\eta_d)}(\boldsymbol{x}^\top\boldsymbol{z}) \tag{36}$$

$$= \frac{1}{|\mathbb{S}^d|}\sum_{n=0}^{N}\frac{(2n+d-1)(n+d-2)!}{n!(d-1)!}\frac{n!(d-2)!}{(n+d-2)!}C_n^{(\eta_d)}(\boldsymbol{x}^\top\boldsymbol{z}) \tag{37}$$

$$= \frac{1}{|\mathbb{S}^d|}\sum_{n=0}^{N}\frac{2n+d-1}{d-1}C_n^{(\eta_d)}(\boldsymbol{x}^\top\boldsymbol{z}) \tag{38}$$

$$= \frac{1}{|\mathbb{S}^d|}\sum_{n=0}^{N}\frac{n+\eta_d}{\eta_d}C_n^{(\eta_d)}(\boldsymbol{x}^\top\boldsymbol{z}) \tag{39}$$

$$= \frac{1}{|\mathbb{S}^d|}\sum_{n=0}^{N}\left[C_n^{(\eta_d+1)}(\boldsymbol{x}^\top\boldsymbol{z}) - C_{n-2}^{(\eta_d+1)}(\boldsymbol{x}^\top\boldsymbol{z})\right] \tag{40}$$

$$= \frac{1}{|\mathbb{S}^d|}\left[C_N^{(\eta_d+1)}(\boldsymbol{x}^\top\boldsymbol{z}) + C_{N-1}^{(\eta_d+1)}(\boldsymbol{x}^\top\boldsymbol{z})\right], \tag{41}$$

where Eq. (40) follows from $C_n^{(\eta_d+1)}(\boldsymbol{x}^\top\boldsymbol{z}) - C_{n-2}^{(\eta_d+1)}(\boldsymbol{x}^\top\boldsymbol{z}) = \frac{n+\eta_d}{\eta_d}C_n^{(\eta_d)}(\boldsymbol{x}^\top\boldsymbol{z})$ (Lemma H.1). For notational simplicity, we define $C_{-1}^{(\eta_d+1)}(\boldsymbol{x}^\top\boldsymbol{z}) = C_{-2}^{(\eta_d+1)}(\boldsymbol{x}^\top\boldsymbol{z}) = 0$ in the above expressions. In addition, when $d = 1$, we can confirm that Eq. (41) equals to $\frac{1}{2\pi}\frac{\sin((N+0.5)\rho(\boldsymbol{x},\boldsymbol{z}))}{\sin(\rho(\boldsymbol{x},\boldsymbol{z})/2)}$ (Lemma H.8), which matches the definition of $b_{N,\boldsymbol{z}}(\boldsymbol{x})$ for $d = 1$. Thus, for any $d \geq 1$, $b_{N,\boldsymbol{z}}(\boldsymbol{x})$ can be written as

$$b_{N,\boldsymbol{z}}(\boldsymbol{x}) = \frac{1}{|\mathbb{S}^d|}\left[C_N^{(\eta_d+1)}(\boldsymbol{x}^\top\boldsymbol{z}) + C_{N-1}^{(\eta_d+1)}(\boldsymbol{x}^\top\boldsymbol{z})\right]. \tag{42}$$

By using the above expression, we show properties 1 and 2.

**Regarding Property 1.** Property 1 follows from the basic known properties of the Gegenbauer polynomial. Firstly, the absolute value of a Gegenbauer polynomial $|C_N^{(\eta)}(t)|$ attains a maximum at $t = 1$ for any $\eta > 0$ (Lemmas H.3 and H.4). Secondly, it is known that $C_N^{(\eta)}(1) = \frac{(n+2\eta-1)!}{n!(2\eta-1)!} > 0$ holds (Lemma H.3). Thus, by noting that $\boldsymbol{z}^\top\boldsymbol{z} = 1$ holds, we immediately obtain $\boldsymbol{z} \in \arg\max_{\boldsymbol{x}\in\mathbb{S}^d} f_{\epsilon,N,\boldsymbol{z}}(\boldsymbol{x})$, $2\epsilon = f_{\epsilon,N,\boldsymbol{z}}(\boldsymbol{z})$, and $\forall\boldsymbol{x} \in \mathbb{S}^d, |f_{\epsilon,N,\boldsymbol{z}}(\boldsymbol{x})| \leq 2\epsilon$.

**Regarding Property 2.** By setting $c = 1$ in Lemma H.10, there exists a constant $C_d > 0$ such that the following inequality holds for any $\boldsymbol{x} \in \{\widetilde{\boldsymbol{x}} \in \mathbb{S}^d \mid N^{-1} \leq \rho(\widetilde{\boldsymbol{x}},\boldsymbol{z}) \leq \pi - N^{-1}\}$:

$$|f_{\epsilon,N,\boldsymbol{z}}(\boldsymbol{x})| \leq \begin{cases} \epsilon C_d N^{\eta_d-d}\rho(\boldsymbol{x},\boldsymbol{z})^{-(\eta_d+1)} & \text{if } \rho(\boldsymbol{x},\boldsymbol{z}) \in [N^{-1},\pi/2], \\ \epsilon C_d N^{\eta_d-d}(\pi-\rho(\boldsymbol{x},\boldsymbol{z}))^{-(\eta_d+1)} & \text{if } \rho(\boldsymbol{x},\boldsymbol{z}) \in [\pi/2,\pi-N^{-1}]. \end{cases} \tag{43}$$

Here, for $\xi = C_d^{1/(\eta_d+1)}N^{-1}$, we have

$$\epsilon C_d N^{\eta_d-d}\xi^{-(\eta_d+1)} = \epsilon N^{2\eta_d-d+1} = \epsilon. \tag{44}$$

Thus, by setting $w_\epsilon^{\text{new}}/2 = \widehat{C}_d N^{-1}$ with a sufficiently large constant such that $\widehat{C}_d \geq \max\{C_d^{1/(\eta_d+1)}, 1\}$, we have $|f_{\epsilon,N,\boldsymbol{z}}(\boldsymbol{x})| \leq \epsilon$ for any $\boldsymbol{x} \in \{\widetilde{\boldsymbol{x}} \in \mathbb{S}^d \mid \rho(\widetilde{\boldsymbol{x}},\boldsymbol{z}) \in [w_\epsilon^{\text{new}}/2, \pi - w_\epsilon^{\text{new}}/2]\}$. This implies that the $\epsilon$-optimal region is a subset of $\{\widetilde{\boldsymbol{x}} \in \mathbb{S}^d \mid \rho(\widetilde{\boldsymbol{x}},\boldsymbol{z}) \in [0, w_\epsilon^{\text{new}}/2) \cup (\pi - w_\epsilon^{\text{new}}/2, \pi]\}$. By noting $\rho(\boldsymbol{x},\boldsymbol{z}) = \pi - \rho(\boldsymbol{x},-\boldsymbol{z})$, we obtain the desired result. $\qquad\square$

### D.3. Proof of Property 3 of Lemma 4.1

*Proof.* From the definition of $b_{N,\boldsymbol{z}}$, the function $f_{\epsilon,N,\boldsymbol{z}}$ is rewritten as follows:

$$f_{\epsilon,N,\boldsymbol{z}}(\boldsymbol{x}) = \sum_{n=0}^{N}\sum_{j=1}^{N_{n,d+1}}\alpha_{n,j}\sqrt{\lambda_n}Y_{n,j}(\boldsymbol{x}), \quad \text{where } \alpha_{n,j} = \frac{2\epsilon Y_{n,j}(\boldsymbol{z})}{b_{N,\boldsymbol{z}}(\boldsymbol{z})\sqrt{\lambda_n}}. \tag{45}$$

Thus, by applying Mercer's representation theorem (Theorem 2.7), we have

$$\|f_{\epsilon,N,\boldsymbol{z}}\|_{\mathcal{H}_k(\mathbb{S}^d)} = \sqrt{\sum_{n=0}^{N}\sum_{j=1}^{N_{n,d+1}}\alpha_{n,j}^2} = \frac{2\epsilon}{b_{N,\boldsymbol{z}}(\boldsymbol{z})}\sqrt{\sum_{n=0}^{N}\sum_{j=1}^{N_{n,d+1}}\left(\frac{Y_{n,j}(\boldsymbol{z})}{\sqrt{\lambda_n}}\right)^2}. \tag{46}$$

Here, from $\lambda_n \geq \underline{\lambda}_n$ and the monotonicity of $\underline{\lambda}_n$, the above equation implies

$$\|f_{\epsilon,N,\boldsymbol{z}}\|_{\mathcal{H}_k(\mathbb{S}^d)} \leq \frac{2\epsilon}{b_{N,\boldsymbol{z}}(\boldsymbol{z})\sqrt{\underline{\lambda}_N}}\sqrt{\sum_{n=0}^{N}\sum_{j=1}^{N_{n,d+1}}Y_{n,j}^2(\boldsymbol{z})} = \frac{2\epsilon}{\sqrt{\underline{\lambda}_N}b_{N,\boldsymbol{z}}(\boldsymbol{z})}. \tag{47}$$

The last equality follows from the definition of $b_{N,\boldsymbol{z}}(\boldsymbol{z}) := \sum_{n=0}^{N}\sum_{j=1}^{N_{n,d+1}}Y_{n,j}^2(\boldsymbol{z})$. From the addition theorem, we further obtain the explicit expression for $b_{N,\boldsymbol{z}}(\boldsymbol{z})$ as follows:

$$b_{N,\boldsymbol{z}}(\boldsymbol{z}) = \sum_{n=0}^{N}\sum_{j=1}^{N_{n,d+1}}Y_{n,j}^2(\boldsymbol{z}) = \frac{1}{|\mathbb{S}^d|}\sum_{n=0}^{N}N_{n,d+1}P_{n,d+1}(\boldsymbol{z}^\top\boldsymbol{z}) = \frac{1}{|\mathbb{S}^d|}\sum_{n=0}^{N}N_{n,d+1}, \tag{48}$$

where the last equality uses $P_{n,d+1}(z^\top z) = P_{n,d+1}(1) = 1$ from the definition of $P_{n,d+1}$. Thus, the RKHS norm $\|f_{\epsilon,N,z}\|_{\mathcal{H}_k(\mathbb{S}^d)}$ satisfies

$$\|f_{\epsilon,N,z}\|_{\mathcal{H}_k(\mathbb{S}^d)} \leq 2\epsilon\sqrt{\frac{|\mathbb{S}^d|}{\underline{\lambda}_N \sum_{n=0}^N N_{n,d+1}}} \leq 2\epsilon\sqrt{\frac{|\mathbb{S}^d|}{\underline{\lambda}_N}}. \tag{49}$$

Then, from the definition of $\underline{\lambda}_N$, there exist constants $A_{d,\theta} > 0$ and $a_{d,\theta} > 0$ such that

$$\|f_{\epsilon,N,z}\|_{\mathcal{H}_k(\mathbb{S}^d)} \leq \epsilon(A_{d,\theta}N)^{a_{d,\theta}N} \tag{50}$$

for any sufficiently large $N$. Here, we set $\bar{N}$ as follows:

$$\bar{N} = \frac{1}{a_{d,\theta}}\left(\ln\frac{B}{\epsilon}\right)\left(\ln\ln\frac{B}{\epsilon}\right)^{-1}. \tag{51}$$

Then, if $B/\epsilon > 1$, we have

$$\epsilon(A_{d,\theta}\bar{N})^{a_{d,\theta}\bar{N}} \leq B \Leftrightarrow a_{d,\theta}\bar{N}\ln(A_{d,\theta}\bar{N}) \leq \ln\frac{B}{\epsilon} \tag{52}$$

$$\Leftrightarrow \left(\ln\frac{B}{\epsilon}\right)\left(\ln\ln\frac{B}{\epsilon}\right)^{-1}\ln\left(\frac{A_{d,\theta}}{a_{d,\theta}}\left(\ln\frac{B}{\epsilon}\right)\left(\ln\ln\frac{B}{\epsilon}\right)^{-1}\right) \leq \ln\frac{B}{\epsilon} \tag{53}$$

$$\Leftrightarrow \left(\ln\ln\frac{B}{\epsilon}\right)^{-1}\left[\left(\ln\ln\frac{B}{\epsilon}\right) + \ln\left(\frac{A_{d,\theta}}{a_{d,\theta}}\left(\ln\ln\frac{B}{\epsilon}\right)^{-1}\right)\right] \leq 1 \tag{54}$$

$$\Leftrightarrow 1 + \frac{\ln\left(\frac{A_{d,\theta}}{a_{d,\theta}}\left(\ln\ln\frac{B}{\epsilon}\right)^{-1}\right)}{\ln\ln\frac{B}{\epsilon}} \leq 1 \tag{55}$$

$$\Leftrightarrow 1 + \frac{\ln\frac{A_{d,\theta}}{a_{d,\theta}} - \ln\ln\ln\frac{B}{\epsilon}}{\ln\ln\frac{B}{\epsilon}} \leq 1. \tag{56}$$

Therefore, if $B/\epsilon$ is sufficiently large such that $B/\epsilon > 1$ and $\ln\frac{A_{d,\theta}}{a_{d,\theta}} - \ln\ln\ln\frac{B}{\epsilon} \leq 1$ hold, we have $\|f_{\epsilon,\bar{N},z}\|_{\mathcal{H}_k(\mathbb{S}^d)} = \|f^{\mathrm{new}}_{z;\epsilon}\|_{\mathcal{H}_k(\mathbb{S}^d)} \leq B$. $\qquad\square$

### D.4. Proof of Lemma 4.3

*Proof.* Fix any $z \in \mathcal{N}^{\mathrm{new}}_\epsilon$. Then, we decompose the summation based on the geodesic distance from $\mathcal{R}^{\mathrm{new}}_{z;\epsilon}$ as follows:

$$\sum_{\tilde{z}\in\mathcal{N}^{\mathrm{new}}_\epsilon}\sup_{x\in\mathcal{R}^{\mathrm{new}}_{z;\epsilon}}|f^{\mathrm{new}}_{\tilde{z};\epsilon}(x)|^2 = \sup_{x\in\mathcal{R}^{\mathrm{new}}_{z;\epsilon}}|f^{\mathrm{new}}_{z;\epsilon}(x)|^2 + \sum_{i\geq 0}\sum_{\tilde{z}\in\mathcal{Z}_i}\sup_{x\in\mathcal{R}^{\mathrm{new}}_{z;\epsilon}}|f^{\mathrm{new}}_{\tilde{z};\epsilon}(x)|^2 \tag{57}$$

where $\mathcal{Z}_i = \{\tilde{z} \in \mathcal{N}^{\mathrm{new}}_\epsilon \mid iw^{\mathrm{new}}_\epsilon/2 < \rho(\mathcal{R}^{\mathrm{new}}_{z;\epsilon},\tilde{z}) \leq (i+1)w^{\mathrm{new}}_\epsilon/2\}$ with $\rho(\mathcal{R}^{\mathrm{new}}_{z;\epsilon},\tilde{z}) := \inf_{x\in\mathcal{R}^{\mathrm{new}}_{z;\epsilon}}\rho(x,\tilde{z})$. The above equality follows from $\mathcal{N}^{\mathrm{new}}_\epsilon = \{z\} \cup \left(\bigcup_{i\geq 0}\mathcal{Z}_i\right)$, which is immediately implied by the definitions of $\mathcal{Z}_i$. Furthermore, from Lemma H.7, we have the following inequalities for some constant $\tilde{C}_d > 0$:

$$\sum_{i\geq 0}\sum_{\tilde{z}\in\mathcal{Z}_i}\sup_{x\in\mathcal{R}^{\mathrm{new}}_{z;\epsilon}}|f^{\mathrm{new}}_{\tilde{z};\epsilon}(x)|^2 \leq \sum_{i\geq 0}|\mathcal{Z}_i|\sup_{\tilde{z}\in\mathcal{Z}_i}\sup_{x\in\mathcal{R}^{\mathrm{new}}_{z;\epsilon}}|f^{\mathrm{new}}_{\tilde{z};\epsilon}(x)|^2 \tag{58}$$

$$\leq \tilde{C}_d\sup_{\tilde{z}\in\mathcal{Z}_0}\sup_{x\in\mathcal{R}^{\mathrm{new}}_{z;\epsilon}}|f^{\mathrm{new}}_{\tilde{z};\epsilon}(x)|^2 + \tilde{C}_d\sum_{i\geq 1}i^{d-1}\sup_{\tilde{z}\in\mathcal{Z}_i}\sup_{x\in\mathcal{R}^{\mathrm{new}}_{z;\epsilon}}|f^{\mathrm{new}}_{\tilde{z};\epsilon}(x)|^2 \tag{59}$$

$$\leq 4\tilde{C}_d\epsilon^2 + \tilde{C}_d\sum_{i\geq 1}i^{d-1}\sup_{\tilde{z}\in\mathcal{Z}_i}\sup_{x\in\mathcal{R}^{\mathrm{new}}_{z;\epsilon}}|f^{\mathrm{new}}_{\tilde{z};\epsilon}(x)|^2. \tag{60}$$

In addition,

$$\widetilde{C}_d \sum_{i\in\mathbb{N}_+} i^{d-1} \sup_{\widetilde{\boldsymbol{z}}\in\mathcal{Z}_i} \sup_{\boldsymbol{x}\in\mathcal{R}^{\text{new}}_{\widetilde{\boldsymbol{z}};\epsilon}} |f^{\text{new}}_{\widetilde{\boldsymbol{z}};\epsilon}(\boldsymbol{x})|^2 \tag{61}$$

$$\leq \widetilde{C}_d \sum_{i\in\mathbb{N}_+} i^{d-1} \sup_{\widetilde{\boldsymbol{z}}\in\mathcal{Z}_i} \sup_{\boldsymbol{x}\in\mathcal{R}^{\text{new}}_{\widetilde{\boldsymbol{z}};\epsilon}} \epsilon^2 C_d \bar{N}^{2(\eta_d-d)} \min\{\rho(\boldsymbol{x},\widetilde{\boldsymbol{z}}), \pi - \rho(\boldsymbol{x},\widetilde{\boldsymbol{z}})\}^{-2(\eta_d+1)} \tag{62}$$

$$= \epsilon^2 \widetilde{C}_d C_d \bar{N}^{2(\eta_d-d)} \sum_{i\in\mathbb{N}_+} i^{d-1} \sup_{\widetilde{\boldsymbol{z}}\in\mathcal{Z}_i} \sup_{\boldsymbol{x}\in\mathcal{R}^{\text{new}}_{\widetilde{\boldsymbol{z}};\epsilon}} \min\{\rho(\boldsymbol{x},\widetilde{\boldsymbol{z}}),\, \rho(-\boldsymbol{x},\widetilde{\boldsymbol{z}})\}^{-2(\eta_d+1)} \tag{63}$$

$$\leq \epsilon^2 \widetilde{C}_d C_d \bar{N}^{2(\eta_d-d)} \sum_{i\in\mathbb{N}_+} i^{d-1} \sup_{\widetilde{\boldsymbol{z}}\in\mathcal{Z}_i} \rho(\mathcal{R}^{\text{new}}_{\boldsymbol{z};\epsilon},\widetilde{\boldsymbol{z}})^{-2(\eta_d+1)} \tag{64}$$

$$\leq \epsilon^2 \widetilde{C}_d C_d \bar{N}^{2(\eta_d-d)} \sum_{i\in\mathbb{N}_+} i^{d-1} \left(i w^{\text{new}}_\epsilon/2\right)^{-2(\eta_d+1)} \tag{65}$$

$$\leq \epsilon^2 \widetilde{C}_d C_d 4^{\eta_d+1} \bar{N}^{2(\eta_d-d)} \left(w^{\text{new}}_\epsilon\right)^{-2(\eta_d+1)} \sum_{i\in\mathbb{N}_+} i^{d-1-2(\eta_d+1)} \tag{66}$$

$$= \epsilon^2 \widetilde{C}_d C_d 4^{\eta_d+1} \bar{N}^{2(\eta_d-d)} \left(2\widehat{C}_d \bar{N}^{-1}\right)^{-2(\eta_d+1)} \sum_{i\in\mathbb{N}_+} i^{-2} \tag{67}$$

$$= \epsilon^2 \widetilde{C}_d C_d 4^{\eta_d+1} \left(2\widehat{C}_d\right)^{-2(\eta_d+1)} \frac{\pi^2}{6}, \tag{68}$$

where:

- Eq. (62) follows from Eq. (43). The constant $C_d$ is the same constant as in Eq. (43) (i.e., the constant used in Lemma H.10 with $c = 1$). Here, note that $\rho(\widetilde{\boldsymbol{z}},\boldsymbol{x}) > w^{\text{new}}_\epsilon/2$ and $\rho(\widetilde{\boldsymbol{z}},-\boldsymbol{x}) > w^{\text{new}}_\epsilon/2$ hold for any $\widetilde{\boldsymbol{z}}\in\mathcal{Z}_i$ and $\boldsymbol{x}\in\mathcal{R}^{\text{new}}_{\widetilde{\boldsymbol{z}};\epsilon}$ with $i\in\mathbb{N}_+$, which are the conditions to apply Eq. (43). In addition, note that $\min\{\rho(\boldsymbol{x},\widetilde{\boldsymbol{z}}), \pi - \rho(\boldsymbol{x},\widetilde{\boldsymbol{z}})\} = \rho(\boldsymbol{x},\widetilde{\boldsymbol{z}})$ if $\rho(\boldsymbol{x},\widetilde{\boldsymbol{z}}) \leq \pi/2$; otherwise, $\min\{\rho(\boldsymbol{x},\widetilde{\boldsymbol{z}}), \pi - \rho(\boldsymbol{x},\widetilde{\boldsymbol{z}})\} = \pi - \rho(\boldsymbol{x},\widetilde{\boldsymbol{z}})$.

- Eq. (64) holds since $\min\{\rho(\boldsymbol{x},\widetilde{\boldsymbol{z}}),\, \rho(-\boldsymbol{x},\widetilde{\boldsymbol{z}})\} \geq \rho(\mathcal{R}^{\text{new}}_{\boldsymbol{z};\epsilon},\widetilde{\boldsymbol{z}})$, which is implied by $\boldsymbol{x}\in\mathcal{R}^{\text{new}}_{\boldsymbol{z};\epsilon}$ and $-\boldsymbol{x}\in\mathcal{R}^{\text{new}}_{\boldsymbol{z};\epsilon}$ from the definition of $\mathcal{R}^{\text{new}}_{\boldsymbol{z};\epsilon}$.

- Eq. (65) follows from the definition of $\mathcal{Z}_i$.

- Eq. (67) follows from the definition of $w^{\text{new}}_\epsilon$. Here, $\widehat{C}_d$ is the constant depending only on $d$. (The constant $\widehat{C}_d$ is defined in the sentence below Eq. (44)).

- Eq. (68) follows from $\sum_{i\in\mathbb{N}_+} i^{-2} = \pi^2/6$ and $\bar{N}^{2(\eta_d-d)+2(\eta_d+1)} = \bar{N}^{4\eta_d-2d+2} = \bar{N}^{2(d-1)-2d+2} = 1$.

Finally, from Eqs. (57), (58) and (68), we have

$$\sum_{\widetilde{\boldsymbol{z}}\in\mathcal{N}^{\text{new}}_\epsilon} \sup_{\boldsymbol{x}\in\mathcal{R}^{\text{new}}_{\widetilde{\boldsymbol{z}};\epsilon}} |f^{\text{new}}_{\widetilde{\boldsymbol{z}};\epsilon}(\boldsymbol{x})|^2 = \sup_{\boldsymbol{x}\in\mathcal{R}^{\text{new}}_{\boldsymbol{z};\epsilon}} |f^{\text{new}}_{\boldsymbol{z};\epsilon}(\boldsymbol{x})|^2 + \sum_{i\geq 0}\sum_{\widetilde{\boldsymbol{z}}\in\mathcal{Z}_i} \sup_{\boldsymbol{x}\in\mathcal{R}^{\text{new}}_{\widetilde{\boldsymbol{z}};\epsilon}} |f^{\text{new}}_{\widetilde{\boldsymbol{z}};\epsilon}(\boldsymbol{x})|^2 \tag{69}$$

$$\leq 4\epsilon^2 + 4\widetilde{C}_d\epsilon^2 + \epsilon^2 \widetilde{C}_d C_d 4^{\eta_d+1} \left(2\widehat{C}_d\right)^{-2(\eta_d+1)} \frac{\pi^2}{6} \tag{70}$$

$$= \epsilon^2 \left[4 + 4\widetilde{C}_d + \widetilde{C}_d C_d 4^{\eta_d+1} \left(2\widehat{C}_d\right)^{-2(\eta_d+1)} \frac{\pi^2}{6}\right]. \tag{71}$$

The above inequality is the desired statement. □

# E. Proofs of Theorems in Section 3

The proofs in this section follow those provided in (Cai & Scarlett, 2021) except for the definition of the hard function used in the proofs.

### E.1. Proof of Theorem 3.1

*Proof.* Fix $\epsilon > 0$ and $\boldsymbol{z} \in \mathcal{N}_\epsilon^{\text{new}}$, and consider any algorithm whose simple regret at step $T$ is at most $\epsilon$ with probability at least $1 - \delta$ for any $f \in \mathcal{H}_k(\mathbb{S}^d, B)$. Assume that $\epsilon$ is sufficiently small such that the properties in Lemma 4.1 hold under the RKHS norm upper bound of $B/3$. Then, for $\widetilde{\boldsymbol{z}} \in \mathcal{N}_\epsilon^{\text{new}}$ with $\widetilde{\boldsymbol{z}} \neq \boldsymbol{z}$, let us define the functions $f$ and $\widetilde{f}$ as $f = f_{\boldsymbol{z};\epsilon}^{\text{new}}$ and $\widetilde{f} = f_{\boldsymbol{z};\epsilon}^{\text{new}} + 2f_{\widetilde{\boldsymbol{z}};\epsilon}^{\text{new}}$, respectively. Here, let $\mathcal{A}$ be the event that the estimated maximizer $\widehat{\boldsymbol{x}}_T$ is in the region $\mathcal{R}_{\boldsymbol{z};\epsilon}^{\text{new}}$ (i.e., $\widehat{\boldsymbol{x}}_T \in \mathcal{R}_{\boldsymbol{z};\epsilon}^{\text{new}}$). Then, if the algorithm attains the simple regret at most $\epsilon$ for both $f$ and $\widetilde{f}$ with probability at least $1 - \delta$, we have $\mathbb{P}_f(\mathcal{A}) \geq 1 - \delta$ and $\mathbb{P}_{\widetilde{f}}(\mathcal{A}) \leq \delta$. Thus, by leveraging Lemma 2.3, we have

$$\sum_{\boldsymbol{x} \in \mathcal{N}_\epsilon^{\text{new}}} \mathbb{E}_f[N_{\boldsymbol{x}}(T)] \overline{D}_{f,\widetilde{f}}^{(\boldsymbol{x})} \geq \ln \frac{1}{2.4\delta}, \tag{72}$$

where $N_{\boldsymbol{x}}(T) := \sum_{t=1}^T \mathbb{1}\{\boldsymbol{x}_t \in \mathcal{R}_{\boldsymbol{x};\epsilon}^{\text{new}}\}$ is the number of query points within $\mathcal{R}_{\boldsymbol{x};\epsilon}^{\text{new}}$ up to step $T$, and

$$\overline{D}_{f,\widetilde{f}}^{(\boldsymbol{x})} := \sup_{\widetilde{\boldsymbol{x}} \in \mathcal{R}_{\boldsymbol{x};\epsilon}^{\text{new}}} \text{KL}\left(\mathcal{N}(f(\widetilde{\boldsymbol{x}}), \sigma^2) \| \mathcal{N}(\widetilde{f}(\widetilde{\boldsymbol{x}}), \sigma^2)\right). \tag{73}$$

Since the KL divergence between two Gaussian measures with common variance is given as $\text{KL}\left(\mathcal{N}(f(\widetilde{\boldsymbol{x}}), \sigma^2) \| \mathcal{N}(\widetilde{f}(\widetilde{\boldsymbol{x}}), \sigma^2)\right) = \frac{(f(\widetilde{\boldsymbol{x}}) - \widetilde{f}(\widetilde{\boldsymbol{x}}))^2}{2\sigma^2}$ (Scarlett et al., 2017), we obtain

$$\overline{D}_{f,\widetilde{f}}^{(\boldsymbol{x})} = \sup_{\widetilde{\boldsymbol{x}} \in \mathcal{R}_{\boldsymbol{x};\epsilon}^{\text{new}}} \frac{2\left(f_{\widetilde{\boldsymbol{z}};\epsilon}^{\text{new}}(\widetilde{\boldsymbol{x}})\right)^2}{\sigma^2}. \tag{74}$$

By combining the above equation of $\overline{D}_{f,\widetilde{f}}^{(\boldsymbol{x})}$ with Eq. (72), we have

$$\frac{2}{\sigma^2} \sum_{\boldsymbol{x} \in \mathcal{N}_\epsilon^{\text{new}}} \mathbb{E}_f[N_{\boldsymbol{x}}(T)] \sup_{\widetilde{\boldsymbol{x}} \in \mathcal{R}_{\boldsymbol{x};\epsilon}^{\text{new}}} \left(f_{\widetilde{\boldsymbol{z}};\epsilon}^{\text{new}}(\widetilde{\boldsymbol{x}})\right)^2 \geq \ln \frac{1}{2.4\delta}. \tag{75}$$

Summing the above inequality for all $\widetilde{\boldsymbol{z}} \neq \boldsymbol{z}$ in $\mathcal{N}_\epsilon^{\text{new}}$, we obtain

$$\frac{2}{\sigma^2} \sum_{\substack{\widetilde{\boldsymbol{z}} \in \mathcal{N}_\epsilon^{\text{new}} \\ \widetilde{\boldsymbol{z}} \neq \boldsymbol{z}}} \sum_{\boldsymbol{x} \in \mathcal{N}_\epsilon^{\text{new}}} \mathbb{E}_f[N_{\boldsymbol{x}}(T)] \sup_{\widetilde{\boldsymbol{x}} \in \mathcal{R}_{\boldsymbol{x};\epsilon}^{\text{new}}} \left(f_{\widetilde{\boldsymbol{z}};\epsilon}^{\text{new}}(\widetilde{\boldsymbol{x}})\right)^2 \geq (M_\epsilon^{\text{new}} - 1) \ln \frac{1}{2.4\delta} \tag{76}$$

$$\Leftrightarrow \frac{2}{\sigma^2} \sum_{\boldsymbol{x} \in \mathcal{N}_\epsilon^{\text{new}}} \mathbb{E}_f[N_{\boldsymbol{x}}(T)] \sum_{\widetilde{\boldsymbol{z}} \in \mathcal{N}_\epsilon^{\text{new}}; \widetilde{\boldsymbol{z}} \neq \boldsymbol{z}} \left(\sup_{\widetilde{\boldsymbol{x}} \in \mathcal{R}_{\boldsymbol{x};\epsilon}^{\text{new}}} \left(f_{\widetilde{\boldsymbol{z}};\epsilon}^{\text{new}}(\widetilde{\boldsymbol{x}})\right)^2\right) \geq (M_\epsilon^{\text{new}} - 1) \ln \frac{1}{2.4\delta} \tag{77}$$

$$\Rightarrow \frac{2C_d \epsilon^2}{\sigma^2} \sum_{\boldsymbol{x} \in \mathcal{N}_\epsilon^{\text{new}}} \mathbb{E}_f[N_{\boldsymbol{x}}(T)] \geq (M_\epsilon^{\text{new}} - 1) \ln \frac{1}{2.4\delta} \tag{78}$$

$$\Leftrightarrow \frac{2C_d T \epsilon^2}{\sigma^2} \geq (M_\epsilon^{\text{new}} - 1) \ln \frac{1}{2.4\delta}, \tag{79}$$

where $C_d > 0$ is some constant depending on $d$. In the above statement, the third line follows from Lemma 4.3, and the last line follows from $\sum_{\boldsymbol{x} \in \mathcal{N}_\epsilon^{\text{new}}} \mathbb{E}_f[N_{\boldsymbol{x}}(T)] = \mathbb{E}_f[\sum_{\boldsymbol{x} \in \mathcal{N}_\epsilon^{\text{new}}} N_{\boldsymbol{x}}(T)] = T$. Since $M_\epsilon^{\text{new}} = \Theta((w_\epsilon^{\text{new}})^{-d}) = \Theta\left((\ln \frac{B}{\epsilon})^d (\ln \ln \frac{B}{\epsilon})^{-d}\right)$, the above inequality implies

$$T \geq \Omega\left(\frac{\sigma^2}{\epsilon^2} \left(\ln \frac{B}{\epsilon}\right)^d \left(\ln \ln \frac{B}{\epsilon}\right)^{-d} \left(\ln \frac{1}{\delta}\right)\right). \tag{80}$$

$\square$

### E.2. Proof of Theorem 3.2

*Proof.* Fix $\epsilon > 0$ and $\boldsymbol{z} \in \mathcal{N}_\epsilon^{\text{new}}$, and consider any algorithm whose cumulative regret at step $T$ is at most $T\epsilon/2$ with probability at least $1 - \delta$ for any $f \in \mathcal{H}_k(\mathbb{S}^d, B)$. Assume that $\epsilon$ is sufficiently small such that the properties in Lemma 4.1 hold with the RKHS norm upper bound of $B/3$. As with the proof of Lemma 3.1, for $\widetilde{\boldsymbol{z}} \in \mathcal{N}_\epsilon^{\text{new}}$ with $\widetilde{\boldsymbol{z}} \neq \boldsymbol{z}$, we define the functions $f$ and $\widetilde{f}$ as $f = f_{\boldsymbol{z};\epsilon}^{\text{new}}$ and $\widetilde{f} = f_{\boldsymbol{z};\epsilon}^{\text{new}} + 2f_{\widetilde{\boldsymbol{z}};\epsilon}^{\text{new}}$, respectively. Here, let $\mathcal{A}$ be the event that the number of query points within the region $\mathcal{R}_{\boldsymbol{z};\epsilon}^{\text{new}}$ is at least $T/2$ (i.e., $\sum_{t=1}^{T} \mathbb{1}\{\boldsymbol{x}_t \in \mathcal{R}_{\boldsymbol{z};\epsilon}^{\text{new}}\} \geq T/2$). Then, due to the condition of the algorithm, we have $\mathbb{P}_f(\mathcal{A}) \geq 1 - \delta$ and $\mathbb{P}_{\widetilde{f}}(\mathcal{A}) \leq \delta$. Thus, by following the same proof strategy of Theorem 3.1, we have

$$T \geq \frac{(M_\epsilon^{\text{new}} - 1)\sigma^2}{2\epsilon^2 C_d} \ln \frac{1}{2.4\delta}, \tag{81}$$

where $C_d > 0$ is some constant depending on $d$. From the above arguments, we observe that if $\mathbb{P}_f(R_T \leq \epsilon T/2) \geq 1 - \delta$ holds for any $f \in \mathcal{H}_k(\mathbb{S}^d, B)$, then the inequality in Eq. (81) must hold. Thus, by considering the contrapositive statement, if the following inequality holds:

$$T < \frac{(M_\epsilon^{\text{new}} - 1)\sigma^2}{2\epsilon^2 C_d} \ln \frac{1}{2.4\delta}, \tag{82}$$

then, the inequality $\mathbb{P}_f(R_T \leq \epsilon T/2) < 1 - \delta$ holds for some $f \in \mathcal{H}_k(\mathbb{S}^d, B)$. This implies that, for any algorithm, if the inequality in Eq. (82) holds, there exists some function $f \in \mathcal{H}_k(\mathbb{S}^d, B)$ such that $\mathbb{P}_f(R_T > \epsilon T/2) = 1 - \mathbb{P}_f(R_T \leq \epsilon T/2) > \delta$. The remaining part of the proof is to adjust $\epsilon$ so that the lower bound becomes as large as possible under the inequality in Eq. (82). Since $M_\epsilon^{\text{new}} = \Theta((w_\epsilon^{\text{new}})^{-d}) = \Theta\left(\left(\ln \frac{B}{\epsilon}\right)^d \left(\ln\ln \frac{B}{\epsilon}\right)^{-d}\right)$, Eq. (82) is implied by

$$\epsilon < \widetilde{C}_{d,\theta} \sqrt{\frac{\sigma^2}{T} \left(\ln \frac{B}{\epsilon}\right)^d \left(\ln\ln \frac{B}{\epsilon}\right)^{-d} \left(\ln \frac{1}{\delta}\right)} \tag{83}$$

for sufficiently small constant $\widetilde{C}_{d,\theta} > 0$. To obtain the desired choice of $\epsilon$ from the above inequality, we follow the same arguments provided in Chapter V in (Scarlett et al., 2017). We analyze the following equation, such that the above inequality is tight up to a factor of $1/2$:

$$\epsilon = \frac{\widetilde{C}_{d,\theta}}{2} \sqrt{\frac{\sigma^2}{T} \left(\ln \frac{B}{\epsilon}\right)^d \left(\ln\ln \frac{B}{\epsilon}\right)^{-d} \left(\ln \frac{1}{\delta}\right)}. \tag{84}$$

The above inequality further implies

$$\frac{\epsilon}{B} = \frac{\widetilde{C}_{d,\theta}}{2} \sqrt{\frac{\sigma^2 \left(\ln \frac{1}{\delta}\right)}{B^2}} \sqrt{\frac{1}{T} \left(\ln \frac{B}{\epsilon}\right)^d \left(\ln\ln \frac{B}{\epsilon}\right)^{-d}}. \tag{85}$$

Thus, we can set $\epsilon/B$ sufficiently small by taking a sufficiently small implied constant in the condition $\sigma^2(\ln \frac{1}{\delta})/B^2 = O(T)$. Next, we study the behavior of the logarithmic terms in the right-hand side of the above equation. From Eq. (84), we have

$$\ln \frac{B}{\epsilon} = \ln \sqrt{\frac{B^2 T}{\sigma^2 (\ln \frac{1}{\delta})}} - \frac{d}{2} \ln \left(\left(\frac{2}{\widetilde{C}_{d,\theta}}\right)^{\frac{2}{d}} \left(\ln \frac{B}{\epsilon}\right) \left(\ln\ln \frac{B}{\epsilon}\right)^{-1}\right). \tag{86}$$

For sufficiently large $B/\epsilon$, the second term on the right-hand side of the above equation satisfies $\frac{d}{2} \ln \left(\left(\frac{2}{\widetilde{C}_{d,\theta}}\right)^{\frac{2}{d}} \left(\ln \frac{B}{\epsilon}\right) \left(\ln\ln \frac{B}{\epsilon}\right)^{-1}\right) \leq \frac{1}{2} \ln \frac{B}{\epsilon}$. Therefore, provided that the implied constant of $\sigma^2(\ln \frac{1}{\delta})/B^2 = O(T)$ is sufficiently small, and Eq. (84) holds, then,

$$\frac{1}{3} \ln \frac{B^2 T}{\sigma^2 (\ln \frac{1}{\delta})} \leq \ln \frac{B}{\epsilon}. \tag{87}$$

Furthermore, the function $t/\ln t$ is non-decreasing for $t > e$. Therefore, if $\sigma^2(\ln \frac{1}{\delta})/B^2 = O(T)$ holds with sufficiently small implied constant, we have $\frac{1}{3} \ln \frac{B^2 T}{\sigma^2 (\ln \frac{1}{\delta})} > e$ and

$$\frac{\widehat{C}_d}{2} \sqrt{\frac{\sigma^2}{T} \left(\ln \frac{B^2 T}{\sigma^2 (\ln \frac{1}{\delta})}\right)^d \left(\ln\ln \frac{B^2 T}{\sigma^2 (\ln \frac{1}{\delta})}\right)^{-d} \left(\ln \frac{1}{\delta}\right)} \leq \epsilon \tag{88}$$

for some constant $\widehat{C}_d > 0$. The above inequality implies that we can take the desired $\epsilon$ such that

$$\epsilon = \Omega\left(\sqrt{\frac{\sigma^2}{T}\left(\ln\frac{B^2 T}{\sigma^2(\ln\frac{1}{\delta})}\right)^d\left(\ln\ln\frac{B^2 T}{\sigma^2(\ln\frac{1}{\delta})}\right)^{-d}\left(\ln\frac{1}{\delta}\right)}\right), \tag{89}$$

which implies the desired cumulative regret lower bound $T\epsilon/2 = \Omega\left(\sqrt{T\sigma^2\left(\ln\frac{B^2 T}{\sigma^2(\ln\frac{1}{\delta})}\right)^d\left(\ln\ln\frac{B^2 T}{\sigma^2(\ln\frac{1}{\delta})}\right)^{-d}\left(\ln\frac{1}{\delta}\right)}\right).$

$\square$

## F. Proof of Theorem 5.1

*Proof.* For any $\mathcal{X} \subset \{x \in \mathbb{R}^d \mid \|x\|_2 \leq 1\}$ or $\mathcal{X} \subset \mathbb{S}^d$, the inequality $\gamma_T(\mathcal{X}) \leq \gamma_T(\mathbb{S}^d)$ holds under the SE kernel from Lemma 9 in (Iwazaki, 2025b). Therefore, it is enough to show that the desired statement is valid for $\gamma_T(\mathbb{S}^d)$[6]. Here, from Lemma 15 in (Iwazaki, 2025b), we have

$$\gamma_T(\mathbb{S}^d) \leq \left(\sum_{n=0}^{M} N_{n,d+1}\right)\ln\left(1 + \frac{T}{\lambda^2}\right) + \frac{T}{|\mathbb{S}^d|\lambda^2}\sum_{n=M+1}^{\infty}\lambda_n N_{n,d+1} \tag{90}$$

for any $M \in \mathbb{N}$. In the above inequality, $\lambda_n$ is the eigenvalue defined in Eq. (8). Although Eq. (8) provides the lower bound, an upper bound of the same order is valid (Theorem 2 in Minh et al., 2006). Namely, we have

$$\forall n \in \mathbb{N}_+, \ \lambda_n \leq \left(\frac{2e}{\theta}\right)^n \frac{|\mathbb{S}^d|\bar{C}_{d,\theta}}{(2n+d-1)^{n+\frac{d}{2}}} \tag{91}$$

for some constant $\bar{C}_{d,\theta} > 0$. By combining Eq. (90) with Eq. (91), we obtain the simple upper bound on the MIG. Specifically, there exist some constants $C_{d,\theta,\lambda^2}, c_{d,\theta} > 0$ such that

$$\gamma_T(\mathbb{S}^d) \leq C_{d,\theta,\lambda^2}\left(M^d \ln T + T\sum_{n=M+1}^{\infty}(c_{d,\theta}n)^{-n}\right) \tag{92}$$

for any $M \in \mathbb{N}_+$ and $T \geq 2$. See Eq. (216) in (Iwazaki, 2025b). In the above inequality, Iwazaki (2025b) choose $M = \Theta(\ln T)$ to obtain the existing $O(\ln^{d+1} T)$ upper bound. However, this result can be further improved by a more careful choice of $M$. First, for $M > 1/c_{d,\theta}$, we have

$$\sum_{n=M+1}^{\infty}(c_{d,\theta}n)^{-n} \leq \sum_{n=M+1}^{\infty}(c_{d,\theta}M)^{-n} \tag{93}$$

$$\leq \int_{M}^{\infty}(c_{d,\theta}M)^{-n}\,\mathrm{d}n \tag{94}$$

$$= \frac{(c_{d,\theta}M)^{-M}}{\ln(c_{d,\theta}M)}. \tag{95}$$

Thus, we have

$$\gamma_T(\mathbb{S}^d) \leq C_{d,\theta,\lambda^2}\left(M^d \ln T + T\frac{(c_{d,\theta}M)^{-M}}{\ln(c_{d,\theta}M)}\right) \tag{96}$$

for any $M > 1/c_{d,\theta}$. Then, to cancel out the effect of $T$ on the second term of the right-hand side of the above inequality, we set $M$ such that $(\ln T) \leq M\ln(c_{d,\theta}M) \leq c_{d,\theta}^{(U)}(\ln T)$. By taking a sufficiently large constant $c_{d,\theta}^{(U)} > 0$, such an $M$ with $M > 1/c_{d,\theta}$ always exists for any $T \geq 2$. Then, we have

$$\ln(c_{d,\theta}M) \geq \ln\ln T - \ln\left(\ln(c_{d,\theta}M)\right). \tag{97}$$

---

[6]Note that we can obtain the same result even for a compact $\mathcal{X}$ not contained in $\{x \in \mathbb{R}^d \mid \|x\|_2 \leq 1\}$ by rescaling the lengthscale parameter $\theta$.

Since $M$ can be chosen as a non-decreasing sequence from the definition, we can take $M$ such that $\ln\left(\ln(c_{d,\theta}M)\right) \le C'_{d,\theta}\ln(c_{d,\theta}M)$ for any $T \ge 2$, where $C'_{d,\theta} > 0$ is some constant. Then, we have

$$\ln(c_{d,\theta}M) \ge \frac{1}{1+C'_{d,\theta}}\ln\ln T > 0 \tag{98}$$

for any $T \ge 3$. Furthermore, from the definition of $M$, we have

$$T\frac{(c_{d,\theta}M)^{-M}}{\ln(c_{d,\theta}M)} \le T(c_{d,\theta}M)^{-\ln_{(c_{d,\theta}M)}T}\frac{(1+C'_{d,\theta})}{\ln\ln T} \tag{99}$$

$$\le \frac{1+C'_{d,\theta}}{\ln\ln T} \tag{100}$$

for any $T \ge 3$. In addition, we have $M \le c^{(U)}_{d,\theta}(\ln T)(\ln(c_{d,\theta}M))^{-1} \le c^{(U)}_{d,\theta}(1+C'_{d,\theta})\frac{\ln T}{\ln\ln T}$. Therefore, from Eq. (96), for any $T \ge 3$, we have

$$\gamma_T(\mathbb{S}^d) \le C_{d,\theta,\lambda^2}\left(\left(c^{(U)}_{d,\theta}(1+C'_{d,\theta})\frac{\ln T}{\ln\ln T}\right)^d(\ln T)+\frac{1+C'_{d,\theta}}{\ln\ln T}\right) = O\left(\frac{(\ln T)^{d+1}}{(\ln\ln T)^d}\right). \tag{101}$$

$\square$

# G. Detailed Discussion of Extension to $\mathcal{X} = [0,1]^d$

We provide the detailed discussion outlined in Section 6. In this section, for simplicity, we assume $d = 1$.

### G.1. Discussion of Proof using Mercer Decomposition under Gaussian Measure

As discussed in Chapter 4.3.1 in (Rasmussen & Williams, 2005), the eigenvalues $\lambda_n^{\mathrm{Gauss}}$ and eigenfunctions $\widehat{\phi}_n^{\mathrm{Gauss}}$ of the integral operator of the SE kernel under the Gaussian measure $\mathcal{N}(0,\sigma^2)$ are given as follows:

$$\lambda_n^{\mathrm{Gauss}} = \sqrt{\frac{2a}{A}}B^n, \tag{102}$$

$$\widehat{\phi}_n^{\mathrm{Gauss}}(x) = \exp(-(c-a)x^2)H_n(\sqrt{2c}x), \tag{103}$$

where $H_n$ is the $n$-th order Hermite polynomial, and $a$, $b$, $c$, $A$, $B$ are defined as $a^{-1} = 4\sigma^2$, $b^{-1} = \theta$, $c = \sqrt{a^2+2ab}$, $A = a+b+c$, and $B = b/A$. Here, note that the above eigenfunctions are not normalized; thus, we use the normalized eigenfunction $\phi_n^{\mathrm{Gauss}}(x)$, which is defined as follows:

$$\phi_n^{\mathrm{Gauss}}(x) = c_n\widehat{\phi}_n^{\mathrm{Gauss}}(x), \tag{104}$$

where $c_n := 1/\sqrt{\int(\widehat{\phi}_n^{\mathrm{Gauss}}(x))^2p_{\mathrm{Gauss}}(x)\mathrm{d}x}$ is the normalizing constant, and $p_{\mathrm{Gauss}}(x)$ is the probability density function for $\mathcal{N}(0,\sigma^2)$. Here, from the basic orthogonality properties of Hermite polynomials (e.g., Chapter 22 in Abramowitz & Stegun, 1968), we can confirm $c_n = 1/\sqrt{\sqrt{\frac{a}{c}}2^n n!}$. As the natural extension of our hard function on $\mathbb{S}^d$, let us consider the following approximated version of delta function $g_{\epsilon,N}(x)$ centered at $0$:

$$g_{\epsilon,N}(x) = \frac{2\epsilon}{b_N(0)}b_N(x), \quad \text{where } b_N(x) = \sum_{n=0}^{N}\phi_n^{\mathrm{Gauss}}(x)\phi_n^{\mathrm{Gauss}}(0). \tag{105}$$

Since $\lambda_n^{\mathrm{Gauss}} = \Theta(\exp(-cn))$ for some constant $c > 0$, we can show that $\|g_{\epsilon,\bar{N}}\|_{\mathcal{H}_k(\mathbb{R})} \le B$ holds with $\bar{N} = \Theta(\ln\frac{B}{\epsilon})$ by relying on Mercer's representation theorem. (We omit the proof, as it follows the same proof strategy as that of property 3 in Lemma 4.1). Furthermore, as with our hard function construction on $\mathbb{S}^d$, the above definition of $b_N$ can be simplified as

follows:

$$\sum_{n=0}^{N} \phi_n^{\text{Gauss}}(x)\phi_n^{\text{Gauss}}(0) = \sum_{n=0}^{N} c_n^2 \exp(-(c-a)x^2)H_n(\sqrt{2c}x)H_n(0) \tag{106}$$

$$= \sqrt{\frac{c}{a}} \exp(-(c-a)x^2) \sum_{n=0}^{N} \frac{H_n(\sqrt{2c}x)H_n(0)}{2^n n!} \tag{107}$$

$$= \sqrt{\frac{c}{a}} \exp(-(c-a)x^2) \frac{1}{2^N N!} \frac{H_N(0)H_{N+1}(\sqrt{2c}x) - H_N(\sqrt{2c}x)H_{N+1}(0)}{\sqrt{2c}x}. \tag{108}$$

In the last line, we use the Christoffel-Darboux formula for Hermite polynomials (Chapter 22 in Abramowitz & Stegun, 1968). Here, if $N$ is odd, it is known that $H_N(0) = 0$ holds; thus, $b_N(x)$ can be simplified as follows:

$$b_N(x) = \begin{cases} -\sqrt{\frac{c}{a}} \exp(-(c-a)x^2)\frac{1}{2^N N!}\frac{H_{N+1}(0)}{\sqrt{2c}x}H_N(\sqrt{2c}x) & \text{if } N \text{ is odd}, \\ \sqrt{\frac{c}{a}} \exp(-(c-a)x^2)\frac{1}{2^N N!}\frac{H_N(0)}{\sqrt{2c}x}H_{N+1}(\sqrt{2c}x) & \text{if } N \text{ is even}. \end{cases} \tag{109}$$

Although we omit the detailed proofs, by combining the above forms with the known properties of the Hermite polynomial, we can obtain the results analogous to properties 1 and 2 in Lemma 2.1. However, the dependence of $\bar{N}$ on $w_\epsilon$ becomes $w_\epsilon = \Theta(1/\sqrt{\bar{N}})$, which leads to $w_\epsilon = \Theta(1/\sqrt{\ln(B/\epsilon)})$. We can confirm this by the fact that $\exp(-x^2/2)H_N(x) = \Theta(\cos(\sqrt{N}x))$ for sufficiently large $N$ (Chapter 8 in Szegö, 1939), which suggests that the width of the function around 0 decreases with $\Theta(1/\sqrt{N})$. Thus, we cannot obtain the improved $w_\epsilon$ by using the function in Eq. (105). This implies that, at least if we follow the existing proof strategy, the resulting lower bound becomes $\Omega(\epsilon^{-2}(\ln(1/\epsilon))^{d/2})$ and $\Omega(\sqrt{T(\ln T)^{d/2}})$ as with the existing results.

### G.2. Discussion of Proof using Mercer Decomposition under Uniform Measure

To our knowledge, unlike the Gaussian measure, no existing literature quantifies the Mercer decomposition under uniform measure on a general compact domain in explicit form. Thus, we need to study eigensystems without resorting to explicit forms. Regarding the eigenvalues, it is known that $\lambda_n^{\text{unif}}$ decreases as $\lambda_n^{\text{unif}} = \Theta((Cn)^{-cn})$ for some constants $C, c > 0$ (Theorem III in Widom, 1964), as with the $\lambda_n$ in Eq. (8). This will be useful for showing the analogous result to property 3 in Lemma 4.1. However, so far, we are not aware of any existing literature that provides useful insight into eigenfunctions $\phi_n^{\text{unif}}$, which will be essential for deriving the analogical results for properties 1 and 2 in Lemmas 4.1 and the upper bound in Lemma 4.3. Thus, we believe that developing new analytical ideas will be essential.

## H. Helper Lemmas

### H.1. Basic Properties of Gegenbauer Polynomial

The Gegenbauer polynomial $C_n^{(\eta)}(x)$, which is also called an ultraspherical polynomial, is an orthogonal polynomial whose weight function is $(1 - x^2)^{\eta - 1/2}$. For the readers who are not familiar with the Gegenbauer polynomial, we refer to (Szegö, 1939) or Chapter 22 in (Abramowitz & Stegun, 1968). Below, we summarize the basic properties of the Gegenbauer polynomial, which are used in our proofs.

**Lemma H.1** (Eq. (4.7.29) in (Szegö, 1939) or Eq. (22.7.23) in (Abramowitz & Stegun, 1965)). *For any $\eta > -1/2$, $n \in \mathbb{N}_+$, and $t \in [-1, 1]$, the following equation holds:*

$$(n + \eta)C_n^{(\eta)}(t) = \eta[C_n^{(\eta+1)}(t) - C_{n-2}^{(\eta+1)}(t)]. \tag{110}$$

*Here, $C_{-1}^{(\eta)}(t)$ is defined as $C_{-1}^{(\eta)}(t) = 0$.*

**Lemma H.2** (Gegenbauer polynomial and Legendre polynomial, Eq. (2.145) in (Atkinson & Han, 2012)). *For any $n \in \mathbb{N}$, $t \in [-1, 1]$, and $d \geq 2$, the following equation holds:*

$$C_n^{((d-1)/2)}(t) = \frac{(n + d - 2)!}{n!(d - 2)!} P_{n,d+1}(t). \tag{111}$$

**Lemma H.3** (Values of Gegenbauer polynomial, Eqs. (4.7.3) and (4.7.4) in (Szegö, 1939)). *For any $n \in \mathbb{N}$, $t \in [-1, 1]$, and $\eta > -1/2$, the following equations hold:*

$$C_n^{(\eta)}(1) = \frac{(n + 2\eta - 1)!}{n!(2\eta - 1)!}, \quad C_n^{(\eta)}(-t) = (-1)^n C_n^{(\eta)}(t). \tag{112}$$

**Lemma H.4** (Maximum of Gegenbauer polynomial, Theorem 7.33.1 in (Szegö, 1939)). *For any $n \in \mathbb{N}$ and $\eta > 0$, the following equation holds:*

$$\max_{t \in [-1,1]} |C_n^{(\eta)}(t)| = \frac{(n + 2\eta - 1)!}{n!(2\eta - 1)!}. \tag{113}$$

**Lemma H.5** (Upper bound on Gegenbauer polynomial, Eq. (7.33.6) in (Szegö, 1939)). *Fix any $c > 0$ and $\eta > 0$. Then, there exists a constant $C_{c,\eta} > 0$ such that the following statement holds:*

$$\forall n \in \mathbb{N}_+, \forall \xi \in [cn^{-1}, \pi/2], \ |C_n^{(\eta)}(\cos(\xi))| \le C_{c,\eta} \xi^{-\eta} n^{\eta-1}. \tag{114}$$

## H.2. Helper Lemmas about Packing Numbers

Lemmas H.6 and H.7 below provide the upper bounds on the packing number of the annular regions on a hypersphere, which are required by our proof (Lemma 4.3) and by the extension of the existing proof (Lemma A.3).

**Lemma H.6.** *Fix any $z \in \mathcal{N}_\epsilon^{(s)}$ and $\epsilon > 0$. Let $\mathcal{N}_\epsilon^{(s)}$ and $\mathcal{R}_{z;\epsilon}^{(s)}$ be the $w_\epsilon$-separated set and the partition of $\mathbb{S}^d$ defined in Definitions A.1 and A.2. Let us define $\mathcal{Z}_i^{(s)}$ as $\mathcal{Z}_i^{(s)} = \{\widetilde{z} \in \mathcal{N}_\epsilon^{(s)} \mid iw_\epsilon/2 < \inf_{x \in \mathcal{R}_{z;\epsilon}^{(s)}} \|x - \widetilde{z}\|_2 \le (i+1)w_\epsilon/2\}$. Then, there exists some constant $C_d > 0$ such that the following inequality holds:*

$$|\mathcal{Z}_i^{(s)}| \le \begin{cases} C_d & \text{if } i = 0, \\ C_d i^{d+1} & \text{if } i \in \mathbb{N}_+. \end{cases} \tag{115}$$

*Proof.* Let $\mathcal{S}_i^{(s)} := \{\widetilde{z} \in \mathbb{S}^d \mid iw_\epsilon/2 < \inf_{x \in \mathcal{R}_{z;\epsilon}^{(s)}} \|x - \widetilde{z}\|_2 \le (i+1)w_\epsilon/2\}$. Then, from the definition of $\mathcal{N}_\epsilon^{(s)}$, we have $|\mathcal{Z}_i^{(s)}| \le P(\mathcal{S}_i^{(s)}, w_\epsilon, \|\cdot\|_\infty)$. To obtain the upper bound of $P(\mathcal{S}_i^{(s)}, w_\epsilon, \|\cdot\|_\infty)$, we also define the set $\bar{\mathcal{S}}_i^{(s)} := \{\bar{z} \in \mathbb{S}^d \mid iw_\epsilon/2 < \|z - \widetilde{z}\|_2 \le (i+1+2\sqrt{d+1})w_\epsilon/2\}$. Here, from the following observations, we conclude that $\mathcal{S}_i^{(s)} \subset \bar{\mathcal{S}}_i^{(s)}$:

- For any $\widetilde{z} \in \mathcal{S}_i^{(s)}$, we have $\|z - \widetilde{z}\|_2 > iw_\epsilon/2$ from the definition of $\mathcal{S}_i^{(s)}$.

- If $\sup_{x \in \mathcal{R}_{z;\epsilon}^{(s)}} \|z - x\|_\infty > w_\epsilon$ holds, there exists $x \in \mathcal{R}_{z;\epsilon}^{(s)}$ such that $\forall \widetilde{z} \in \mathcal{N}_\epsilon^{(s)}$, $\|\widetilde{z} - x\|_\infty > w_\epsilon$ holds from the definition of $\mathcal{R}_{z;\epsilon}^{(s)}$. However, this implies that we can add a closed ball of radius $w_\epsilon/2$ centered at $x$ to $\mathcal{N}_\epsilon^{(s)}$ without breaking the assumption of $w_\epsilon$-separated set. This contradicts the definition of the packing number. Thus, $\sup_{x \in \mathcal{R}_{z;\epsilon}^{(s)}} \|z - x\|_\infty \le w_\epsilon$. By using this, for any $\widetilde{z} \in \mathcal{S}_i^{(s)}$, we have $\|z - \widetilde{z}\|_2 \le \sup_{x \in \mathcal{R}_{z;\epsilon}^{(s)}} \|z - x\|_2 + (i+1)w_\epsilon/2 \le \sqrt{d+1} \sup_{x \in \mathcal{R}_{z;\epsilon}^{(s)}} \|z - x\|_\infty + (i+1)w_\epsilon/2 \le (i+1+2\sqrt{d+1})w_\epsilon/2$.

Therefore, we have $P(\mathcal{S}_i^{(s)}, w_\epsilon, \|\cdot\|_\infty) \le P(\bar{\mathcal{S}}_i^{(s)}, w_\epsilon, \|\cdot\|_\infty)$. Furthermore, $\bar{\mathcal{S}}_i^{(s)} \subset \widetilde{\mathcal{S}}_i^{(s)} := \{\widetilde{z} \in \mathbb{R}^{d+1} \mid \|z - \widetilde{z}\|_2 \le (i+1+2\sqrt{d+1})w_\epsilon/2\}$. Note that, with respect to L2-norm, the $w$-packing number of an L2-ball with radius $r$ in $\mathbb{R}^d$ is $\Theta((r/w)^d)$. By combining this with the equivalence of $\|\cdot\|_\infty$ and $\|\cdot\|_2$, we have $P(\widetilde{\mathcal{S}}_i^{(s)}, w_\epsilon, \|\cdot\|_\infty) = \Theta(((i+1+2\sqrt{d+1})/2)^{d+1}) = \Theta(i^{d+1})$, where the hidden constant may depend on $d$. $\square$

**Lemma H.7.** *Fix any $z \in \mathcal{N}_\epsilon^{\text{new}}$ and $\epsilon > 0$. Let $\mathcal{N}_\epsilon^{\text{new}}$ and $\mathcal{R}_{z;\epsilon}^{\text{new}}$ be the $w_\epsilon^{\text{new}}$-separated set and the partition of $\mathbb{S}^d$ defined in Definitions C.1 and C.2. Let us define $\mathcal{Z}_i$ as $\mathcal{Z}_i = \{\widetilde{z} \in \mathcal{N}_\epsilon^{\text{new}} \mid iw_\epsilon^{\text{new}}/2 < \inf_{x \in \mathcal{R}_{z;\epsilon}^{\text{new}}} \rho(x, \widetilde{z}) \le (i+1)w_\epsilon^{\text{new}}/2\}$. Assume that $w_\epsilon^{\text{new}} \le \pi$ holds. Then, there exists some constant $C_d > 0$ such that the following inequality holds:*

$$|\mathcal{Z}_i| \le \begin{cases} C_d & \text{if } i = 0, \\ C_d i^{d-1} & \text{if } i \in \mathbb{N}_+. \end{cases} \tag{116}$$

*Proof.* Let $\mathcal{S}_i := \{\widetilde{z} \in \mathbb{S}^d \mid iw_\epsilon^{\text{new}}/2 < \inf_{x \in \mathcal{R}_{z;\epsilon}^{\text{new}}} \rho(x, \widetilde{z}) \leq (i+1)w_\epsilon^{\text{new}}/2\}$. Then, from the definition of $\mathcal{N}_\epsilon^{\text{new}}$, we have $|\mathcal{Z}_i| \leq P(\mathcal{S}_i, w_\epsilon^{\text{new}}, \rho)$. As with the proof of Lemma H.6, we also define the set $\bar{\mathcal{S}}_i := \{\widetilde{z} \in \mathbb{S}^d \mid iw_\epsilon^{\text{new}}/2 < \rho(z, \widetilde{z}) \leq (i+3)w_\epsilon^{\text{new}}/2\}$. Here, from the following observations, we find $\mathcal{S}_i \subset \bar{\mathcal{S}}_i$:

- For any $\widetilde{z} \in \mathcal{S}_i$, we have $\rho(\widetilde{z}, z) > iw_\epsilon^{\text{new}}/2$ from the definition of $\mathcal{S}_i$.

- If $\sup_{x \in \mathcal{R}_{z;\epsilon}^{\text{new}}} \rho(z, x) > w_\epsilon^{\text{new}}$ holds, there exists $x \in \mathcal{R}_{z;\epsilon}^{\text{new}}$ such that $\forall z \in \mathcal{N}_\epsilon^{\text{new}}$, $\rho(z, x) > w_\epsilon^{\text{new}}$ holds from the definition of $\mathcal{R}_{z;\epsilon}^{\text{new}}$. However, this implies that we can add $w_\epsilon^{\text{new}}/2$-ball centered at $x$ to $\mathcal{N}_\epsilon^{\text{new}}$ without breaking the assumption of $w_\epsilon^{\text{new}}$-separated set. This contradicts the definition of packing number. Thus, $\sup_{x \in \mathcal{R}_{z;\epsilon}^{\text{new}}} \rho(z, x) \leq w_\epsilon^{\text{new}}$. By using this, for any $\widetilde{z} \in \mathcal{S}_i$, we have $\rho(z, \widetilde{z}) \leq \sup_{x \in \mathcal{R}_{z;\epsilon}^{(s)}} \rho(z, x) + (i+1)w_\epsilon/2 \leq (i+3)w_\epsilon^{\text{new}}/2$.

Therefore, we have $P(\mathcal{S}_i, w_\epsilon^{\text{new}}, \rho) \leq P(\bar{\mathcal{S}}_i, w_\epsilon^{\text{new}}, \rho)$. Here, by noting that $P(\bar{\mathcal{S}}_i, w_\epsilon^{\text{new}}, \rho)$ is the maximum number of disjoint closed balls with radius $w_\epsilon^{\text{new}}/2$, its upper bound is given by $V(B_1)/V(B_2)$, where $V(B_1)$ is the surface area of $B_1 \subset \mathbb{S}^d$. Furthermore, $B_1 \subset \mathbb{S}^d$ and $B_2 \subset \mathbb{S}^d$ are defined as $B_1 = \{\widetilde{z} \in \mathbb{S}^d \mid (i-1)w_\epsilon^{\text{new}}/2 < \rho(z, \widetilde{z}) \leq (i+4)w_\epsilon^{\text{new}}/2\}$ and $B_2 = \{\widetilde{z} \in \mathbb{S}^d \mid \rho(z, \widetilde{z}) \leq w_\epsilon^{\text{new}}/2\}$, respectively (see, e.g., Proposition 4.2.12 in Vershynin, 2018).[7] Here, note that the surface area of a spherical cap $\widetilde{B} = \{\widetilde{z} \in \mathbb{S}^d \mid \rho(z, \widetilde{z}) \leq r \leq \pi\}$ is given as $V(\widetilde{B}) = \frac{\pi^{d/2}}{\Gamma(d/2)} \int_0^r \sin^{d-1}(\theta)\mathrm{d}\theta$; thus, we have

$$|\mathcal{Z}_i| \leq \frac{V(B_1)}{V(B_2)} = \frac{\int_{\max\{\min\{(i-1)w_\epsilon^{\text{new}}/2,\pi\},0\}}^{\min\{(i+4)w_\epsilon^{\text{new}}/2,\pi\}} \sin^{d-1}(\theta)\mathrm{d}\theta}{\int_0^{w_\epsilon^{\text{new}}/2} \sin^{d-1}(\theta)\mathrm{d}\theta}. \tag{117}$$

Here, $\sin\theta \geq 2\theta/\pi$ holds for $\theta \in [0, \pi/2]$. Furthermore, $\sin\theta \leq \theta$ holds for any $\theta > 0$. Thus, by noting $w_\epsilon^{\text{new}} \leq \pi$, we have

$$|\mathcal{Z}_i| \leq \frac{\int_{\max\{\min\{(i-1)w_\epsilon^{\text{new}}/2,\pi\},0\}}^{\min\{(i+4)w_\epsilon^{\text{new}}/2,\pi\}} \theta^{d-1}\mathrm{d}\theta}{\int_0^{w_\epsilon^{\text{new}}/2} \left(\frac{2\theta}{\pi}\right)^{d-1}\mathrm{d}\theta} = \left(\frac{\pi}{2}\right)^{d-1} \frac{\int_{\max\{\min\{(i-1)w_\epsilon^{\text{new}}/2,\pi\},0\}}^{\min\{(i+4)w_\epsilon^{\text{new}}/2,\pi\}} \theta^{d-1}\mathrm{d}\theta}{\int_0^{w_\epsilon^{\text{new}}/2} \theta^{d-1}\mathrm{d}\theta}. \tag{118}$$

By simplifying the above inequality, we have

$$|\mathcal{Z}_i| \leq \left(\frac{\pi}{2}\right)^{d-1} \frac{\min\{(i+4)w_\epsilon^{\text{new}}/2,\pi\}^d - \max\{\min\{(i-1)w_\epsilon^{\text{new}}/2,\pi\},0\}^d}{(w_\epsilon^{\text{new}})^d/2^d}. \tag{119}$$

We consider the upper bound on the right-hand side of the above inequality separately based on $i$.

**When $i = 0$.** If $i = 0$, we have

$$|\mathcal{Z}_0| \leq \left(\frac{\pi}{2}\right)^{d-1} \frac{(2w_\epsilon^{\text{new}})^d}{(w_\epsilon^{\text{new}})^d/2^d} = \left(\frac{\pi}{2}\right)^{d-1} 4^d. \tag{120}$$

**When $(i-1)w_\epsilon^{\text{new}}/2 \leq \pi$ and $i \geq 1$.** If $(i-1)w_\epsilon^{\text{new}}/2 \leq \pi$ and $i \geq 1$, we have

$$|\mathcal{Z}_i| \leq \left(\frac{\pi}{2}\right)^{d-1} \left[(i+4)^d - (i-1)^d\right]. \tag{121}$$

By applying the mean-value theorem for the function $x^d$ in the above inequality, we have

$$|\mathcal{Z}_i| \leq \left(\frac{\pi}{2}\right)^{d-1} 5d(i+4)^{d-1} \leq \left(\frac{\pi}{2}\right)^{d-1} 5^d di^{d-1}, \tag{122}$$

where the last inequality follows from $\forall i \geq 1$, $(i+4)^{d-1} \leq (5i)^{d-1}$.

**When $(i-1)w_\epsilon^{\text{new}}/2 \geq \pi$.** If $(i-1)w_\epsilon^{\text{new}}/2 \geq \pi$, we have

$$|\mathcal{Z}_i| \leq 0. \tag{123}$$

Thus, from Eqs. (120), (122), and (123), we obtain the desired statement. $\square$

---

[7]Although the original statement in (Vershynin, 2018) assumes the L2-ball and the standard volume, the result and proof can be easily adapted to a ball with respect to geodesic distance and surface area.

## H.3. Helper Lemmas for $b_{N,\boldsymbol{z}}$ and $f_{\epsilon,N,\boldsymbol{z}}$

The following lemmas, which we leverage in our proofs, are useful for establishing the properties of our new hard functions.

**Lemma H.8** (Gagenbauer polynomials and Dirichlet kernel). *For any $\xi \in (0, \pi]$ and $N \in \mathbb{N}_+$, we have*

$$C_N^{(1)}(\cos(\xi)) + C_{N-1}^{(1)}(\cos(\xi)) = \frac{\sin\left(\left(N + \frac{1}{2}\right)\xi\right)}{\sin\left(\frac{\xi}{2}\right)}. \tag{124}$$

*Proof.* It is known that $C_N^{(1)}(\cos(\xi))$ is equal to the Tchebichef polynomials of the second kind $U_N(\cos(\xi)) := \sin((N + 1)\xi)/\sin(\xi)$ (e.g., Chapter 4.7 in Szegö, 1939). Thus, we have

$$C_N^{(1)}(\cos(\xi)) + C_{N-1}^{(1)}(\cos(\xi)) = \frac{\sin((N+1)\xi)}{\sin(\xi)} + \frac{\sin(N\xi)}{\sin(\xi)} = \frac{2\sin((N+1/2)\xi)\cos(\xi/2)}{\sin(\xi)}, \tag{125}$$

where the last equality uses $\sin(A) + \sin(B) = 2\sin((A + B)/2)\cos((A - B)/2)$. Furthermore, since $\sin(A) = 2\sin(A/2)\cos(A/2)$, we have

$$\frac{2\sin((N+1/2)\xi)\cos(\xi/2)}{\sin(\xi)} = \frac{\sin((N+1/2)\xi)}{\sin(\xi/2)}. \tag{126}$$

The above equation is the desired result. $\qquad\square$

**Lemma H.9.** *For any $N \in \mathbb{N}$, $d \in \mathbb{N}_+$, and $\boldsymbol{z} \in \mathbb{S}^d$, the following inequality holds:*

$$b_{N,\boldsymbol{z}}(\boldsymbol{z}) \geq \frac{N^d}{|\mathbb{S}^d|d!}. \tag{127}$$

*Proof.* It is known that $\sum_{n=0}^{N} N_{n,d+1} = N_{N,d+2}$ from the generating function of the sequence $(N_{n,d})$ (Eq. (2.14) in Atkinson & Han, 2012). From the addition theorem and $P_{n,d+1}(1) = 1$, we have

$$b_{N,\boldsymbol{z}}(\boldsymbol{z}) = \frac{1}{|\mathbb{S}^d|} \sum_{n=0}^{N} N_{n,d+1} = \frac{N_{N,d+2}}{|\mathbb{S}^d|}. \tag{128}$$

Here, from the definition of $N_{n,d+2}$, we have

$$N_{n,d+2} = \frac{2N + d}{d!} \frac{(N + d - 1)!}{N!} \tag{129}$$

$$= \frac{2N + d}{d!} \prod_{m=1}^{d-1} (N + m) \tag{130}$$

$$\geq \frac{N^d}{d!}. \tag{131}$$

The above inequality implies the desired statement. $\qquad\square$

**Lemma H.10.** *Fix any $c > 0$ and $d \in \mathbb{N}_+$. Then, there exists a constant $C_{c,d} > 0$ such that, for any $\boldsymbol{z} \in \mathbb{S}^d$, $N \geq 2$, and $\epsilon > 0$, the following statement holds:*

$$\forall \boldsymbol{x} \in \{\widetilde{\boldsymbol{x}} \in \mathbb{S}^d \mid cN^{-1} \leq \rho(\widetilde{\boldsymbol{x}}, \boldsymbol{z}) \leq \pi - cN^{-1}\}, \tag{132}$$

$$|f_{\epsilon,N,\boldsymbol{z}}(\boldsymbol{x})| \leq \begin{cases} \epsilon C_{c,d} N^{\eta_d - d} \rho(\boldsymbol{x}, \boldsymbol{z})^{-(\eta_d + 1)} & \text{if } \rho(\boldsymbol{x}, \boldsymbol{z}) \in [cN^{-1}, \pi/2], \\ \epsilon C_{c,d} N^{\eta_d - d} (\pi - \rho(\boldsymbol{x}, \boldsymbol{z}))^{-(\eta_d + 1)} & \text{if } \rho(\boldsymbol{x}, \boldsymbol{z}) \in [\pi/2, \pi - cN^{-1}], \end{cases} \tag{133}$$

*where $\eta_d = (d - 1)/2$. Here, $C_{c,d}$ depends only on $c$ and $d$.*

*Proof.* From Eq. (42), we have

$$f_{\epsilon,N,\boldsymbol{z}}(\boldsymbol{x}) = \frac{2\epsilon}{|\mathbb{S}^d|b_{N,\boldsymbol{z}}(\boldsymbol{z})} \left[ C_N^{(\eta_d+1)}(\boldsymbol{x}^\top \boldsymbol{z}) + C_{N-1}^{(\eta_d+1)}(\boldsymbol{x}^\top \boldsymbol{z}) \right]. \tag{134}$$

Using Lemma H.9, we have

$$|f_{\epsilon,N,\boldsymbol{z}}(\boldsymbol{x})| \leq \frac{2\epsilon d!}{N^d} \left[ |C_N^{(\eta_d+1)}(\boldsymbol{x}^\top \boldsymbol{z})| + |C_{N-1}^{(\eta_d+1)}(\boldsymbol{x}^\top \boldsymbol{z})| \right] \tag{135}$$

$$= \frac{2\epsilon d!}{N^d} \left[ |C_N^{(\eta_d+1)}(\cos(\rho(\boldsymbol{x},\boldsymbol{z})))| + |C_{N-1}^{(\eta_d+1)}(\cos(\rho(\boldsymbol{x},\boldsymbol{z})))| \right]. \tag{136}$$

By leveraging Lemma H.5, if $\rho(\boldsymbol{x},\boldsymbol{z}) \in [cN^{-1}, \pi/2]$, the following inequality holds for some constant $\widetilde{C}_{c,d}$:

$$|f_{\epsilon,N,\boldsymbol{z}}(\boldsymbol{x})| \leq \frac{2\epsilon d!}{N^d} \left[ \widetilde{C}_{c,d}\rho(\boldsymbol{x},\boldsymbol{z})^{-(\eta_d+1)}N^{\eta_d} + \widetilde{C}_{c,d}\rho(\boldsymbol{x},\boldsymbol{z})^{-(\eta_d+1)}(N-1)^{\eta_d} \right] \tag{137}$$

$$\leq 4\epsilon d!\widetilde{C}_{c,d}N^{\eta_d-d}\rho(\boldsymbol{x},\boldsymbol{z})^{-(\eta_d+1)}. \tag{138}$$

Furthermore, if $\rho(\boldsymbol{x},\boldsymbol{z}) \in [\pi/2, \pi - cN^{-1}]$, the following inequality follows from Lemma H.3:

$$|f_{\epsilon,N,\boldsymbol{z}}(\boldsymbol{x})| \leq \frac{2\epsilon d!}{N^d} \left[ |C_N^{(\eta_d+1)}(\cos(\rho(\boldsymbol{x},\boldsymbol{z})))| + |C_{N-1}^{(\eta_d+1)}(\cos(\rho(\boldsymbol{x},\boldsymbol{z})))| \right] \tag{139}$$

$$= \frac{2\epsilon d!}{N^d} \left[ |C_N^{(\eta_d+1)}(\cos(\pi - \rho(\boldsymbol{x},\boldsymbol{z})))| + |C_{N-1}^{(\eta_d+1)}(\cos(\pi - \rho(\boldsymbol{x},\boldsymbol{z})))| \right] \tag{140}$$

$$\leq \frac{2\epsilon d!}{N^d} \left[ \widetilde{C}_{c,d} (\pi - \rho(\boldsymbol{x},\boldsymbol{z}))^{-(\eta_d+1)} N^{\eta_d} + \widetilde{C}_{c,d} (\pi - \rho(\boldsymbol{x},\boldsymbol{z}))^{-(\eta_d+1)} (N-1)^{\eta_d} \right] \tag{141}$$

$$\leq 4\epsilon d!\widetilde{C}_{c,d}N^{\eta_d-d}(\pi - \rho(\boldsymbol{x},\boldsymbol{z}))^{-(\eta_d+1)}. \tag{142}$$

Thus, we have

$$|f_{\epsilon,N,\boldsymbol{z}}(\boldsymbol{x})| \leq \begin{cases} 4\epsilon d!\widetilde{C}_{c,d}N^{\eta_d-d}\rho(\boldsymbol{x},\boldsymbol{z})^{-(\eta_d+1)} & \text{if } \rho(\boldsymbol{x},\boldsymbol{z}) \in [cN^{-1}, \pi/2], \\ 4\epsilon d!\widetilde{C}_{c,d}N^{\eta_d-d}(\pi - \rho(\boldsymbol{x},\boldsymbol{z}))^{-(\eta_d+1)} & \text{if } \rho(\boldsymbol{x},\boldsymbol{z}) \in [\pi/2, \pi - cN^{-1}]. \end{cases} \tag{143}$$

Finally, by setting the constant $C_{c,d}$ as $C_{c,d} = 4d!\widetilde{C}_{c,d}$, we obtain the desired statement. $\qquad \square$

