# OpenReview forum: "Tighter Regret Lower Bound for Gaussian Process Bandits with Squared Exponential Kernel in Hypersphere"
_ICML.cc/2026/Conference — ICML 2026 regular_

### Official Review · Reviewer_Nx8e · 2026-03-03

**Soundness:** 3
**Presentation:** 3
**Significance:** 4
**Originality:** 4
**Overall Recommendation:** 5
**Confidence:** 4

**Summary:**

Paper considers the problem of lower-bounding GP for actions sampled from a $ d$-dimensional unit sphere. The novelty is a new lower bound that settles a factor that is exponential in the ambient dimension. The idea used in obtainiing the lower can also be naturally extended to deriving a better information gain bound although the improvement is more modest $\mathcal{O}(-\ln \ln T^d)$.

**Compliance With Llm Reviewing Policy:**

Affirmed.

**Final Justification:**

I found the paper's contributions to be relevant and interesting and I vote for its acceptance.

**Key Questions For Authors:**

I don't have any clarity questions for the authors, as I found the paper to be quite easy to read and follow. I do have one question about the possible extension of the paper:

1. Authors suggest that for extending the ideas to the general domain (say, unit hypercube), the idea of using the eigenfunctions and values looks promising. Wouldn't this have the same issues as, say, trying to improve the lower bound for Matern kernels (I get that the improvement is constant wrt d)? Basically, the eigenfunctions are unknown with respect to the uniform measure, and the parametric form derived in this paper(on a sphere) is quite important. Do authors believe they can "extract" the necessary properties of the eigenfunction by just the PDE boundary conditions? I think this might be quite interesting.

**Limitations:**

Yes

**Strengths And Weaknesses:**

Soundness: I found the ideas in the paper to be clear and well formulated. In particular,  using a projection of the Dirac delta onto the harmonic base to design the hard-function family makes a lot of sense, given the desired properties previously motivated by Scarlett et al., (2017). Overall, the approach is reasonable and appears correct, up to calculation errors.

Presentation: I found the presentation quite good, especially given the paper's theoretical nature. The significance of shaving a $\mathcal{O}(\ln T^{d/4})$ is well motivated and quite different from "usual" kernels where poly-logarithmic terms are often disregarded. Presenting a short summary of results from Scarlett et al.,(2017) is also appreciated and makes the difficulty of the problem more apparent, as well as motivating the author's own approach to designing a hard family instance.
I would ve liked to have seen the argument of deriving the new information gain bound more fleshed out in the main text. I suppose intuitively, given that we have the enttire eigen- function , value system, the usual approach in Vakili et al. (2021)(the boundedness of the eigen functions is of no issue here) should apply, but I don't quite see the more precise treatment of the eigen-decay as noted by the authors.

Significance: I found the paper significant in both its results and its new techniques. The lower bound improvement is significant and the idea of using a harmonic polynomial as the basis in this context is also new. I guess it brings us no closer to closing the  $\ln T^{d/4}$ gap for general domains, but I don't find that to be a big issue, as the sphere is quite standardly used as an action space.

Originality: As noted above, the paper is the first to use a harmonic polynomial basis in the context of designing a hard function class for GP bandits. In my mind, this is sufficient novelty.

---

> ### Author Rebuttal · Authors · 2026-03-27
>
> Thank you for taking the time to review our paper and for your overall positive assessment.
>
> **Q1.** Basically, the eigenfunctions are unknown with respect to the uniform measure, and the parametric form derived in this paper(on a sphere) is quite important. Do authors believe they can "extract" the necessary properties of the eigenfunction by just the PDE boundary conditions? I think this might be quite interesting.
>
> **A1.**
> To our knowledge, some existing works have characterized eigenfunctions under the uniform measure on specific compact domains. For instance, for the Matérn kernel, it is known that the eigenfunctions can be represented as linear combinations of exponential and trigonometric functions (see Section 4.4 of Janz, 2022). Although these specific results for the Matérn kernel do not directly translate to the SE kernel, the underlying analytical techniques may provide a pathway for characterizing its eigenfunctions.
>
> Furthermore, in our framework, the core requirements for the proof are:
>
> - The concentration rate of the projected delta function around its peak (Property 2 in Lemma 4.1).
> - The overall scaling of the function's amplitude (Lemma 4.3).
>
> One promising approach to analyze these properties would be to study the convergence rate of the projected delta function toward the true Dirac delta function in the $L_2$ sense, and then to study the link between this convergence rate and the aforementioned properties. This approach may allow for a proof that bypasses the need for explicit expressions of the eigenfunctions, which we look forward to exploring in future work.
>
> - Janz, D. Sequential decision making with feature-linear models. PhD thesis, 2022.

---

> > ### Author Rebuttal · Reviewer_Nx8e · 2026-04-01
> >
> > I had no major issues with the paper overall, and I vote for its acceptance.

---

### Official Review · Reviewer_LeWp · 2026-03-11

**Soundness:** 3
**Presentation:** 3
**Significance:** 2
**Originality:** 3
**Overall Recommendation:** 4
**Confidence:** 4

**Summary:**

The paper considers the problem of Gaussian Process Bandits on the hypersphere. The primary focus on the paper is to derive tighter lower bounds on the regret of any algorithm for the Squared Exponential kernel where the domain is a hypersphere. The authors use a novel ''hard'' instance over the hypersphere and demonstrate that their construction improves the lower bound by a factor of nearly $(\ln T)^{d/4}$. The authors also provide an improved bound on maximum information gain of SE kernels.

**Compliance With Llm Reviewing Policy:**

Affirmed.

**Key Questions For Authors:**

I have some questions:

1. The most obvious follow up question is that whether such an improvement is also possible for Matern kernels? The paper focuses only on SE kernels (which I think might be too specific).

2. As I mentioned earlier, the analysis technique is interesting. If I understand correctly, the approach in the paper and by the one in Scarlett et al, are two sides of the same coin. The key idea is always to have delta function as your hard function. However, it is not a "nice" function so it is important to approximate it. Scarlett et al considers a "smoothed" version of it (which is not truncated) and you consider an approximation by truncation of Fourier (or an equivalent expansion) series. As we know, both approaches have their unique pros and cons. My question is it possible to combine them to extend the analysis for more general domains, with some geometric structure. Extension to arbitrary compact domains is most likely very hard (and probably not even worth the effort).

3. What about using bijective maps ( probably need them to be diffeomorphic) to extend these argument to more general domains?

4. Can you provide more details on the proof of Theorem 5.1? In particular, can you explain how you avoid that L_{\infty} bound of the eigenfunctions in Eq.(90). It says to refer to Lemma 15 in Iwazaki, 2025b but I could not find any Lemma 15 in the paper.

**Limitations:**

Yes.

**Strengths And Weaknesses:**

I think the strongest points of the paper is its detailed analysis and the construction of the novel hard instance for the hypersphere. Even though the authors have focused on the hypersphere, the recipe looks interesting in general. These ideas can be a useful tool for future studies.

The improvement in this paper is on the lower order terms, which might not be of interest to the community at large. I understand there is dimension dependence in the log terms, and hence it can't be really pushed under the rug (or tilde for that matter). But the matter of fact is that dependence usually not critical to understanding algorithm performance because there is a dominant $\sqrt{T}$ factor.

---

> ### Author Rebuttal · Authors · 2026-03-27
>
> Thank you for taking the time to review our paper and for the positive assessment.
>
> **Q1.** Is improvement also possible for Matern kernels?
>
> **A1.**  While our analytical framework is applicable to the Matérn kernel, it does not yield a tighter lower bound in that case. Our investigation reveals that the improvement for the SE kernel stems from a significant gap between the decay rates of the continuous spectrum (spectral density) and the discrete spectrum (Mercer eigenvalues). For instance, when $d=1$, the spectral density $p(x)$ of the SE kernel is Gaussian, $p(x) = \Theta(\exp(-x^2))$, whereas the eigenvalues $\lambda_n$ decay more slowly as $\lambda_n = \Theta(n^{-n}) = \Theta(\exp(-n \log n))$ (Eq.(8)). This slower decay of eigenvalues allows an improved lower bound compared to the results in (Scarlett et al., 2017), which rely on $p(x)$ in their proof. In contrast, for Matérn kernels, the spectral density and eigenvalues decay at the same polynomial order (e.g., $\Theta(x^{-2\nu-1})$ and $\Theta(n^{-2\nu-1})$ for $d=1$), resulting in no such improvement.
>
> **Q2.** Is it possible to combine them to extend the analysis for more general domains, with some geometric structure?
>
> **A2.**  An advantage of our hard-function construction is its applicability to any Mercer kernel beyond stationary kernels on $\mathbb{R}^d$. However, a challenge remains: deriving necessary properties (as in Lemmas 4.1 and 4.3) requires specific knowledge of the eigensystem, whereas their counterparts are more easily derived in the framework of (Scarlett et al., 2017). Developing a general methodology that bridges these trade-offs for other specific geometric domains beyond the hypersphere would be a highly valuable contribution. We consider this an important and interesting direction for future work.
>
> **Q3.** What about using bijective maps (probably need them to be diffeomorphic)?
>
> **A3.**  It is an insightful question. As the reviewer pointed out, if we can construct a "favorable" bijective map $m: S \to X$ from a spherical subset $S \subset \mathbb{S}^d$ to a general domain $X \subset \mathbb{R}^d$, we could potentially extend our improved lower bound to $X$. Specifically, such a map would need to satisfy:
>
> 1. $m$ is approximately isometric.
> 2. The RKHS norm of the projected hard function, $f_{\text{proj}}(x) = f_{\epsilon, z}^{\text{new}}(m^{-1}(x))$, remains bounded by the original norm: $\||f_{\text{proj}}\||_k = O(\||f\_{\epsilon, z}^{\text{new}}\||\_k)$.
>
> However, establishing Property 2 rigorously is non-trivial. While a diffeomorphism may be included in necessary conditions, to our knowledge, there is little existing literature providing sufficient conditions for the preservation of RKHS norms across geometrically distinct domains. Thus, while this approach has high potential, it requires further non-trivial investigation.
>
> **Q4.** It says to refer to Lemma 15 in Iwazaki, 2025b but I could not find any Lemma 15.
>
> **A4.**
> We have verified that the citation to Lemma 15 in (Iwazaki, 2025**b**) is correct. We suspect the reviewer might have referred to (Iwazaki, 2025**a**). From Lemma 15 in (Iwazaki, 2025b), we can avoid relying on the uniform boundedness of eigenfunctions by utilizing the addition theorem (Eq.(7) in the main paper, or Lemma 14 in Iwazaki, 2025b). Specifically, we can exploit the fact that the tail behavior of the eigenfunctions can be upper-bounded if the "aggregated" eigenfunction values at any fixed point remain bounded.
>
>
> **Additional remark**
> We would like to respectfully remark about the reviewer's following comments:
>
> > The improvement in this paper is on the lower-order terms, which might not be of interest to the community at large. The matter of fact is that dependence is usually not critical to understanding algorithm performance because there is a dominant $\sqrt{T}$ factor.
>
> We agree that for Matérn kernels, the polynomial term is of primary interest as it strictly exceeds $O(\sqrt{T})$. However, for the SE kernel, the dimension-dependent logarithmic factors are highly significant, as they characterize the intrinsic difficulty of the problem relative to simpler instances, such as linear bandits (where the lower bound is $\Omega(d\sqrt{T})$). If we *only* consider the $\sqrt{T}$ term, for a fixed $d$, these problem instances (linear bandits vs kernel bandits with SE kernel) would appear identical in a worst-case asymptotic sense; however, this is clearly wrong. Indeed, to our knowledge, most of the existing literature accounts for the dimension-dependent logarithmic factor by treating it via $O(\sqrt{\gamma_T})$ in its presentation, or use $\tilde{O}(\cdot)$ notation only when considering the kernel class with worse polynomial dependence than $O(\sqrt{T})$ (like Mat\'ern kernel). Therefore, we believe that the dimension-dependent logarithmic factor in the regret bounds for the SE kernel is an important topic in the GP bandit community for correctly understanding the algorithm's performance.

---

> > ### Author Rebuttal · Reviewer_LeWp · 2026-04-03
> >
> > The rebuttal addresses most of my concerns. Unfortunately, I still cannot find Lemma 15 in the cited paper. Based on the citation, this is the link to the paper: https://proceedings.mlr.press/v258/iwazaki25a.html, which does not have a Lemma 15.

---

> > > ### Author Response · Authors · 2026-04-03
> > >
> > > We appreciate your reply. The citation the reviewer is referring to is **(Iwazaki & Takeno, 2025b)**, which is not **(Iwazaki 2025b)**.  Please recheck Lemma 15 in Appendix B of (Iwazaki 2025b) (the following reference):
> > >
> > > - Iwazaki, Shogo. "Improved Regret Bounds for Gaussian Process Upper Confidence Bound in Bayesian Optimization." The Thirty-ninth Annual Conference on Neural Information Processing Systems.
> > >
> > > Again, thank you for taking the time to review our paper.

---

### Official Review · Reviewer_S6re · 2026-03-13

**Soundness:** 3
**Presentation:** 3
**Significance:** 3
**Originality:** 3
**Overall Recommendation:** 4
**Confidence:** 2

**Summary:**

This paper studies sharper upper and lower bounds of cumulative regret bounds for the squared exponential (SE) kernels. The authors provide a tighter regret lower bound that matches the known upper bound in the case when the action space is the $d$-dimensional hypersphere utilizing explicit eigenfunctions of the Mercer's integral operator. Moreover, the authors improve the upper bound of MIG for a general compact subset of the Euclidean space or hypersphere.

**Compliance With Llm Reviewing Policy:**

Affirmed.

**Final Justification:**

The author response resolved my questions, and I will keep my positive assessment.

**Key Questions For Authors:**

- In Section 6, the authors try to extend the result to other compact domains. Do the authors believe that Theorem 3.2 holds for general compact domains?
- The hypersphere (and Euclidean space) are examples of a symmetric space on which harmonic analysis is well-studied. Do the authors suppose the analysis can be extended to other symmetric spaces?

**Limitations:**

Yes

**Strengths And Weaknesses:**

## Strengths
- Even though the action set is limited to the hypersphere, the improved lower bound matches the known upper bound.
- The paper is clearly written. They also include some basic results for spherical harmonics on the hypersphere.
- The second main result holds for any compact subset of the Euclidean space.

## Weaknesses
- As the authors mentioned in the limitations paragraph, the result regarding the improved lower bound is limited to the hypersphere. Since the hypersphere has a very special domain and the proof relies on explicit computation of spherical harmonics, it is unclear that the results can be extended to other compact domains.

---

> ### Author Rebuttal · Authors · 2026-03-27
>
> Thank you for taking the time to review our paper and for your positive assessment.
>
> **Q1.** In Section 6, the authors try to extend the result to other compact domains. Do the authors believe that Theorem 3.2 holds for general compact domains?
>
> **A1.**
> Yes, although it is informal conjecture so far, we believe that the lower bound for a compact domain in $\mathbb{R}^d$ is also $\Omega(\sqrt{T(\log T)^d (\log \log T)^{-d}})$. To be more specific, as detailed in Section 6 and Appendix G.2, we conjecture that the same result can be obtained by extending our hard function construction in Section 4.1 with a uniform measure (or some other bounded support measure). Please also refer to **A1.** for Reviewer Nx8e for details.
>
> **Q2.** The hypersphere (and Euclidean space) are examples of a symmetric space on which harmonic analysis is well-studied. Do the authors suppose the analysis can be extended to other symmetric spaces?
>
> **A2.**
> It is an insightful question. As the reviewer noted, the eigensystems of integral operators on symmetric spaces under shift-invariant measures are often well-studied in harmonic analysis. We believe our proof strategy (summarized in Section 4) is potentially applicable to such spaces, provided that the kernel's eigensystem yields analytically tractable eigenfunctions. Specifically, if one can derive counterparts to our Lemma 4.1 and Lemma 4.3 for a given symmetric space, our framework for establishing lower bounds should naturally follow.

---

> > ### Author Rebuttal · Reviewer_S6re · 2026-04-02
> >
> > I appreciate the responses that resolved my questions, and I will keep my positive assessment

---

### Official Review · Reviewer_nz5d · 2026-03-13

**Soundness:** 4
**Presentation:** 4
**Significance:** 3
**Originality:** 3
**Overall Recommendation:** 5
**Confidence:** 1

**Summary:**

A technically novel and significant work that nearly closes the regret gap for SE-kernel bandits on the hypersphere.

**Compliance With Llm Reviewing Policy:**

Affirmed.

**Final Justification:**

I will keep the same score.

**Key Questions For Authors:**

Since the impact statement mentions potential societal influence, providing a few concrete examples would help clarify the potential impacts.

**Strengths And Weaknesses:**

The contribution is mathematically meaningful and appears novel. While the result does not fully settle the problem on general compact domains, it makes substantial progress on an important special case and introduces a proof technique that could influence future work.

---

> ### Author Rebuttal · Authors · 2026-03-27
>
> Thank you for your overall positive assessment. We are grateful for your effort in reviewing our paper.

---

> > ### Author Rebuttal · Reviewer_nz5d · 2026-04-03
> >
> > My concerns have been adequately addressed.

---

### Decision · Program_Chairs · 2026-04-30

**Decision:**

Accept (regular)

**Comment:**

The paper studies algorithm-independent lower bounds for Gaussian process bandits with the squared exponential kernel, focusing on hyperspherical domains. It provides a tighter cumulative regret lower bound that nearly matches existing upper bounds up to dimension-independent logarithmic factors, and introduces an improved bound on the maximum information gain. The analysis relies on a novel hard-instance construction using harmonic expansions.

Reviewers broadly agree that the paper is technically strong, clearly written, and makes a meaningful theoretical contribution. A key strength is the new lower bound technique, which is seen as both novel and potentially useful beyond the specific setting. The main concern is the restriction to hyperspherical domains and the reliance on specific spectral properties, which may limit generality. Some reviewers also questioned the practical importance of improving logarithmic factors, though others argued this is central for SE kernels.

The rebuttal addressed most technical questions, including clarification of proof details and cited results, and reviewers converged towards acceptance. Overall, the paper makes a solid and nontrivial contribution to the theory of GP bandits, and I recommend acceptance.